# Beta-CROWN: Efficient Bound Propagation with Per-neuron Split Constraints for Neural Network Robustness Verification

Shiqi Wang [* 1]   Huan Zhang [* 2]   Kaidi Xu [* 3]   Xue Lin [3]   Suman Jana [1]   Cho-Jui Hsieh [4]   Zico Kolter [2]

## Abstract

We develop $\beta$-CROWN, a new bound propagation based method that can fully encode neuron split constraints in branch-and-bound (BaB) based complete verification via optimizable parameters $\beta$. When jointly optimized in intermediate layers, $\beta$-CROWN generally produces better bounds than typical LP verifiers with neuron split constraints, while being as efficient and parallelizable as existing bound propagation methods such as CROWN on GPUs. Applied to complete robustness verification benchmarks, $\beta$-CROWN with BaB is close to three orders of magnitude faster than LP-based BaB methods, and is at least 3 times faster than winners of VNN-COMP 2020 competition while producing lower timeout rates. By terminating BaB early, our method can also be used for efficient incomplete verification. We achieve higher verified accuracy in many settings over powerful incomplete verifiers, including those based on convex barrier breaking techniques. Compared to the typically tightest but very costly semidefinite programming (SDP) based incomplete verifiers, we obtain higher verified accuracy with three orders of magnitudes less verification time, and enable better certification for verification-agnostic (e.g., adversarially trained) networks.

## 1. Introduction

As neural networks (NNs) are being deployed in safety-critical applications, it becomes increasingly important to formally verify their behaviors under potentially malicious inputs. Broadly speaking, the neural network verification problem involves proving certain desired relationships between inputs and outputs (often referred to as *specifications*),

*Equal contribution   [1]Columbia University   [2]CMU [3]Northeastern University   [4]UCLA. Correspondence to: Shiqi Wang <sw3215@columbia.edu>, Huan Zhang <huan@huanzhang.com>, Kaidi Xu <xu.kaid@northeastern.edu>.

*Accepted by the ICML 2021 workshop on A Blessing in Disguise: The Prospects and Perils of Adversarial Machine Learning.* Copyright 2021 by the author(s).

such as safety or robustness guarantees, for all inputs inside some domain. This is a challenging problem due to the non-convexity and high dimensionality of neural networks.

We first focus on *complete* verification: the verifier should give a definite "yes/no" answer given sufficient time. Many complete verifiers rely on branch and bound (BaB) (Bunel et al., 2018) involving (1) branching by recursively splitting the original verification problem into subdomains (e.g., splitting a ReLU into positive/negative linear regions) and (2) bounding each subdomain with specialized incomplete verifiers. Traditional BaB-based verifiers use expensive linear programming (LP) solvers (Ehlers, 2017; Lu & Kumar, 2020; Bunel et al., 2020b) as incomplete verifiers which can fully encode neuron split constraints for tight bounding. Meanwhile, a recent work (Xu et al., 2021) shows that cheap incomplete verifiers can significantly accelerate complete verifiers on GPUs over LP-based ones thanks to their efficiency. We call these cheap incomplete verifiers *bound propagation methods* (Wong & Kolter, 2018; Dvijotham et al., 2018; Gehr et al., 2018; Singh et al., 2019b). However, unlike LP solvers, these methods lack the power to handle neuron split constraints in BaB. Such a problem causes loose bounds and unnecessary branching, hurting the verification efficiency. Even worse, they fail to detect many infeasible subdomains in BaB, leading to incompleteness unless additional checking is performed (Xu et al., 2021).

In our work, we develop a new, fast bound propagation based incomplete verifier, $\beta$-CROWN. It solves an optimization problem equivalent to the expensive LP based methods with neuron split constraints while still enjoying the efficiency of bound propagation methods. $\beta$-CROWN contains optimizable parameters $\alpha$ and $\beta$, and any valid settings of these parameters yield sound bounds for verification. These parameters are optimized using a few steps of (super)gradient ascent to achieve bounds as tight as possible. Furthermore, we can jointly optimize $\alpha$ and $\beta$ for intermediate layer bounds, allowing $\beta$-CROWN to tighten relaxations and outperform typical LP verifiers with fixed intermediate layer bounds. Unlike traditional LP-based BaB methods, $\beta$-CROWN can be efficiently implemented with an automatic differentiation framework on GPUs to fully exploit the power of modern accelerators. The combination of $\beta$-CROWN and BaB ($\beta$-CROWN BaB) produces a

complete verifier with GPU acceleration, reducing the verification time of traditional LP based BaB verifiers (Bunel et al., 2018) by almost *three orders of magnitudes* on standard benchmarking models. Compared to state-of-the-art verifiers like ERAN and OVAL (the leading methods in VNN competition 2020 (Liu & Johnson)), our approach is 3 to 70 times faster with similar or lower timeout rates.

Finally, by terminating our complete verifier $\beta$-CROWN BaB early, our approach can also function as a more accurate incomplete verifier by returning an incomplete but sound lower bound of all subdomains explored so far. We achieve better verified accuracy on a few benchmarking models over powerful incomplete verifiers including those based on multi-neuron convex relaxations (Singh et al., 2019a; Tjandraatmadja et al., 2020; Müller et al., 2021) and semidefinite relaxations (Dathathri et al., 2020). Compared to the typically tightest but very costly incomplete verifier SDP-FO (Dathathri et al., 2020) based on tight semidefinite programming (SDP) relaxations (Raghunathan et al., 2018; Dvijotham et al., 2020), our method obtains consistently higher verified accuracy while reducing verification time by three orders of magnitudes, enabling better certification on verification-agnostic (e.g., adversarially trained) models.

## 2. Related Work

Early complete verifiers relied on existing solvers (Katz et al., 2017; Ehlers, 2017; Huang et al., 2017; Dutta et al., 2018; Tjeng et al., 2019) and were limited to very small problem instances. BaB based method was proposed to better exploit the network structure using LP-based incomplete verifier for bounding and ReLU splits for branching (Bunel et al., 2018; Wang et al., 2018a; Lu & Kumar, 2020; Botoeva et al., 2020). Input domain branching was also considered in (Wang et al., 2018b; Royo et al., 2019; Anderson et al., 2019) but limited by input dimensions (Bunel et al., 2018). Traditional complete verifiers may need hours to verify one example on networks with a few thousand neurons. Recently, a few approaches used efficient iterative solvers or bound propagation methods on GPUs without relying on LPs. Bunel et al. (2020a) decomposed the verification problem layer by layer, solved each layer in a closed form on GPUs, and used Lagrangian to enforce consistency between layers. De Palma et al. (2021a) used a dual-space verifier with a tighter linear relaxation (Anderson et al., 2020; Tjandraatmadja et al., 2020) than LP at a cost of high complexity. A concurrent work BaDNB (De Palma et al., 2021b) proposed a new branching strategy "filtered smart branching" for better verification performance. Xu et al. (2020) used CROWN as an efficient incomplete solver on GPUs for complete verification, but it cannot handle neuron split constraints, leading to suboptimal efficiency and more timeouts.

For incomplete verification, Salman et al. (2019) showed the inherent limitation of using per-neuron convex relaxations.

Singh et al. (2019a) and Müller et al. (2021) broke this barrier by relaxing multiple ReLU neurons; Tjandraatmadja et al. (2020) considered relaxing a layer of neurons using a strong mixed-integer programming formulation (Anderson et al., 2019). SDP based relaxations (Raghunathan et al., 2018; Fazlyab et al., 2020; Dvijotham et al., 2020; Dathathri et al., 2020) typically produce tight bounds but with significantly higher cost. We break this barrier using branch and bound, and outperform existing incomplete verifiers under many scenarios in both runtime and tightness.

## 3. $\beta$-CROWN for NN Verification

**Notations.** We define the input of a neural network as $x \in \mathbb{R}^{d_0}$, and define the weights and biases of an $L$-layer neural network as $\mathbf{W}^{(i)} \in \mathbb{R}^{d_i \times d_{i-1}}$ and $\mathbf{b}^{(i)} \in \mathbb{R}^{d_i}$ ($i \in \{1, \cdots, L\}$) respectively. For simplicity we assume that $d_L = 1$ so $\mathbf{W}^{(L)}$ is a vector and $\mathbf{b}^{(L)}$ is a scalar. The neural network function $f : \mathbb{R}^{d_0} \to \mathbb{R}$ is defined as $f(x) = z^{(L)}(x)$, where $z^{(i)}(x) = \mathbf{W}^{(i)}\hat{z}^{(i-1)}(x) + \mathbf{b}^{(i)}$, $\hat{z}^{(i)}(x) = \sigma(z^{(i)}(x))$ and $\hat{z}^{(0)}(x) = x$. $\sigma$ is the activation function and we use ReLU throughout this paper. When the context is clear, we omit $\cdot(x)$ and use $z_j^{(i)}$ and $\hat{z}_j^{(i)}$ to represent the *pre-activation* and *post-activation* values of the $j$-th neuron in the $i$-th layer. The set $\mathcal{C}$ defines the allowed input region. We consider the canonical specification $f(x) > 0$: if we can prove that $f(x) > 0$, $\forall x \in \mathcal{C}$, we say $f(x)$ is verified. Under $x \in \mathcal{C}$, we define the intermediate layer bounds $\mathbf{l}_j^{(i)} \leq z_j^{(i)} \leq \mathbf{u}_j^{(i)}$. When using branch-and-bound, we denote the $\mathcal{Z}^{+(i)}$ and $\mathcal{Z}^{-(i)}$ as the set of neuron indices with positive and negative split constraints in layer $i$. We define the split constraints at layer $i$ as $\mathcal{Z}^{(i)} := \{z^{(i)} \mid z_{j_1}^{(i)} \geq 0, z_{j_2}^{(i)} < 0, \forall j_1 \in \mathcal{Z}^{+(i)}, \forall j_2 \in \mathcal{Z}^{-(i)}\}$. We denote the vector of all pre-ReLU neurons as $z$, and we define a set $\mathcal{Z}$ to represent the split constraints on $z$: $\mathcal{Z} = \mathcal{Z}^{(1)} \cap \mathcal{Z}^{(2)} \cap \cdots \cap \mathcal{Z}^{(L-1)}$. For convenience, we also use the shorthand $\tilde{\mathcal{Z}}^{(i)} := \mathcal{Z}^{(1)} \cap \cdots \cap \mathcal{Z}^{(i)}$ and $\tilde{z}^{(i)} := \{z^{(1)}, z^{(2)}, \cdots, z^{(i)}\}$.

**Bound Propagation with Neuron Split Constraints** The NN verification problem under neuron split constraints can be written as an optimization: $\min_{x \in \mathcal{C}, z \in \mathcal{Z}} f(x)$. Bound propagation methods such as CROWN (Zhang et al., 2018) can give a relatively tight lower bound for $\min_{x \in \mathcal{C}} f(x)$ but they *cannot handle the neuron split constraints* $z \in \mathcal{Z}$. We now show the intuition on how to apply split constraints to the bound propagation process. To encode the neuron splits, we define diagonal matrix $\mathbf{S}^{(i)} \in \mathbb{R}^{d_i \times d_i}$ where $i \in [1, \cdots L-1], j \in [1, \cdots, d_i]$:

$$\mathbf{S}_{j,j}^{(i)} = -1 \ (z_j^{(i)} \geq 0); \ \mathbf{S}_{j,j}^{(i)} = 1 \ (z_j^{(i)} < 0); \ \mathbf{S}_{j,j}^{(i)} = 0 \ (\text{no split})$$

We start from last layer and derive linear bounds for each intermediate layer $z^{(i)}$ and $\hat{z}^{(i)}$ for both $x \in \mathcal{C}$ and $z \in \mathcal{Z}$. We also assume that pre-ReLU bounds $\mathbf{l}^{(i)} \leq z^{(i)} \leq \mathbf{u}^{(i)}$

for each layer $i$ are available. By definition of NN we have:

$$\min_{x \in \mathcal{C}, z \in \mathcal{Z}} f(x) = \min_{x \in \mathcal{C}, z \in \mathcal{Z}} \mathbf{W}^{(L)} \hat{z}^{(L-1)} + \mathbf{b}^{(L)}. \quad (1)$$

Since $\hat{z}^{(L-1)} = \text{ReLU}(z^{(L-1)})$, we can relax ReLU similarly as done in CROWN (see Lemma A.1) at layer $L-1$, and obtain a linear lower bound for $f(x)$ w.r.t. $z^{(L-1)}$:

$$\min_{x \in \mathcal{C}, z \in \mathcal{Z}} f(x) \geq \min_{x \in \mathcal{C}, z \in \mathcal{Z}} \mathbf{W}^{(L)} \mathbf{D}^{(L-1)} z^{(L-1)} + \text{const}.$$

To enforce the split neurons at layer $L-1$, we use a Lagrange function with $\boldsymbol{\beta}^{(L-1)\top} \mathbf{S}^{(L-1)}$ multiplied on $z^{(L-1)}$:

$$\begin{aligned} \min_{\substack{x \in \mathcal{C} \\ z \in \mathcal{Z}}} f(x) &\geq \min_{\substack{x \in \mathcal{C} \\ \tilde{z}^{(L-2)} \in \tilde{\mathcal{Z}}^{(L-2)}}} \max_{\boldsymbol{\beta}^{(L-1)} \geq 0} \mathbf{W}^{(L)} \mathbf{D}^{(L-1)} z^{(L-1)} \\ &+ \boldsymbol{\beta}^{(L-1)\top} \mathbf{S}^{(L-1)} z^{(L-1)} + \text{const} \\ &\geq \max_{\boldsymbol{\beta}^{(L-1)} \geq 0} \min_{\substack{x \in \mathcal{C} \\ \tilde{z}^{(L-2)} \in \tilde{\mathcal{Z}}^{(L-2)}}} \Big( \mathbf{W}^{(L)} \mathbf{D}^{(L-1)} \\ &+ \boldsymbol{\beta}^{(L-1)\top} \mathbf{S}^{(L-1)} \Big) z^{(L-1)} + \text{const} \end{aligned} \quad (2)$$

The first inequality is due to the definition of the Lagrange function: we remove the constraint $z^{(L-1)} \in \mathcal{Z}^{(L-1)}$ and use a multiplier to replace this constraint. The second inequality is due to weak duality. Due to the design of $\mathbf{S}^{(L-1)}$, neuron split $z_j^{(L-1)} \geq 0$ has a negative multiplier $-\boldsymbol{\beta}_j^{(L-1)}$ and split $z_j^{(L-1)} < 0$ has a positive multiplier $\boldsymbol{\beta}_j^{(L-1)}$. Any $\boldsymbol{\beta}^{(L-1)} \geq 0$ yields a lower bound for the constrained optimization. Then we can substitute $z^{(L-1)}$ with $\mathbf{W}^{(L-1)} \hat{z}^{(L-2)} + \mathbf{b}^{(L-1)}$ into Eq. 9 for the next layer. The above bound propagation process continues until reaching input layer, with each layer's split constraints handled via optimizable $\boldsymbol{\beta}$ variables. Following this procedure, we give a linear lower bound for $f(x)$ under split constraints $z \in \mathcal{Z}$:

**Theorem 3.1** ($\boldsymbol{\beta}$**-CROWN bound**). *Given an $L$-layer NN $f(x) : \mathbb{R}^{d_0} \to \mathbb{R}$ with weights $\mathbf{W}^{(i)}$, biases $\mathbf{b}^{(i)}$, pre-ReLU bounds $\mathbf{l}^{(i)} \leq z^{(i)} \leq \mathbf{u}^{(i)}$ ($1 \leq i \leq L$), input bounds $\mathcal{C}$, split constraints $\mathcal{Z}$. We have:*

$$\min_{x \in \mathcal{C}, z \in \mathcal{Z}} f(x) \geq \max_{\boldsymbol{\beta} \geq 0} \min_{x \in \mathcal{C}} (\boldsymbol{a} + \mathbf{P}\boldsymbol{\beta})^\top x + \mathbf{q}^\top \boldsymbol{\beta} + c, \quad (3)$$

*where $\boldsymbol{a} \in \mathbb{R}^{d_0}, \mathbf{P} \in \mathbb{R}^{d_0 \times (\sum_{i=1}^{L-1} d_i)}, \mathbf{q} \in \mathbb{R}^{\sum_{i=1}^{L-1} d_i}$ and $c \in \mathbb{R}$ are functions of $\mathbf{W}^{(i)}$, $\mathbf{b}^{(i)}$, $\mathbf{l}^{(i)}$, $\mathbf{u}^{(i)}$. $\boldsymbol{\beta} := \big[ \boldsymbol{\beta}^{(1)\top} \boldsymbol{\beta}^{(2)\top} \cdots \boldsymbol{\beta}^{(L-1)\top} \big]^\top$ concatenates all $\boldsymbol{\beta}^{(i)}$.*

Detailed formulations for $\boldsymbol{a}$, $\mathbf{P}$, $\mathbf{q}$ and $c$ are given in Appendix C. For $\ell_p$ norm robustness ($\mathcal{C} = \{x \mid \|x - x_0\|_p \leq \epsilon\}$), the inner minimization has a closed solution:

$$\begin{aligned} \min_{x \in \mathcal{C}, z \in \mathcal{Z}} f(x) &\geq \max_{\boldsymbol{\beta} \geq 0} -\|\boldsymbol{a} + \mathbf{P}\boldsymbol{\beta}\|_q \epsilon + (\mathbf{P}^\top x_0 \\ &+ \mathbf{q})^\top \boldsymbol{\beta} + \boldsymbol{a}^\top x_0 + c := \max_{\boldsymbol{\beta} \geq 0} g(\boldsymbol{\beta}) \end{aligned}$$

where $\frac{1}{p} + \frac{1}{q} = 1$. The maximization is concave in $\boldsymbol{\beta}$ ($q \geq 1$), so we can optimize it using projected (super)gradient ascent

with gradients from automatic differentiation. Since any $\boldsymbol{\beta} \geq 0$ yields a valid lower bound for $\min_{x \in \mathcal{C}, z \in \mathcal{Z}} f(x)$, convergence is not necessary to guarantee soundness. An additional optiomizable variable during bound propagation are the slopes of lower bound used for relaxing unstable ReLU neurons (Xu et al., 2021). We define $\boldsymbol{\alpha}^{(i)} \in \mathbb{R}^{d_i}$ as the slopes of lower bounds for unstable ReLU (detailed in Lemma A.1) for layer $i$ and define all free variables $\boldsymbol{\alpha} := \{\boldsymbol{\alpha}^{(1)} \cdots \boldsymbol{\alpha}^{(L-1)}\}$. Since any $0 \leq \boldsymbol{\alpha}_j^{(i)} \leq 1$ yields a valid bound, we can optimize it to tighten the bound. Formally, we rewrite the optimization problem with $\boldsymbol{\alpha}$ explicitly:

$$\min_{x \in \mathcal{C}, z \in \mathcal{Z}} f(x) \geq \max_{0 \leq \boldsymbol{\alpha} \leq 1, \boldsymbol{\beta} \geq 0} g(\boldsymbol{\alpha}, \boldsymbol{\beta}). \quad (4)$$

$g$ is a function of $\mathbf{l}_j^{(i)}$ and $\mathbf{u}_j^{(i)}$ (intermediate layer bounds). These bounds are also computed using $\beta$-CROWN using independent sets of $\boldsymbol{\alpha}, \boldsymbol{\beta}$ variables and are optimized jointly. The joint optimization allows us to outperform typical LP verifier with fixed intermediate bounds (Xu et al., 2021).

**Connections to the Dual Problem** Many BaB based complete verifiers (Bunel et al., 2018; Lu & Kumar, 2020) use an LP formulation as the incomplete verifier (detailed in Section B.2). We first show that it is possible to derive Theorem B.1 from the dual of this LP, leading to Theorem B.2:

**Theorem 3.2.** *The objective $d_{LP}$ for the dual problem of the LP in (Bunel et al., 2018) can be represented as*

$$d_{LP} = -\|\boldsymbol{a} + \mathbf{P}\boldsymbol{\beta}\|_1 \cdot \epsilon + (\mathbf{P}^\top x_0 + \mathbf{q})^\top \boldsymbol{\beta} + \boldsymbol{a}^\top x_0 + c,$$

*where $\boldsymbol{a}$, $\mathbf{P}$, $\mathbf{q}$ and $c$ are defined in the same way as in Theorem B.1, and $\boldsymbol{\beta} \geq 0$ corresponds to the dual variables of neuron split constraints in the LP problem.*

An immediate consequence is that $\beta$-CROWN can potentially solve the verification problem with split constraints as well as using an LP solver with fixed intermediate bounds:

**Corollary 3.2.1.** *When $\boldsymbol{\alpha}$ and $\boldsymbol{\beta}$ are optimally set and intermediate bounds $\mathbf{l}, \mathbf{u}$ are fixed, $\beta$-CROWN produces $p_{LP}^*$, the optimal objective of LP with split constraints in Eq. 14:*

$$\max_{0 \leq \boldsymbol{\alpha} \leq 1, \boldsymbol{\beta} \geq 0} g(\boldsymbol{\alpha}, \boldsymbol{\beta}) = p_{LP}^*,$$

$\beta$**-CROWN with Branch and Bound ($\beta$-CROWN BaB)** We perform complete verification following BaB framework (Bunel et al., 2018) using $\beta$-CROWN as the incomplete solver, and we use simple branching heuristics like BaBSR (Bunel et al., 2020b) or FSB (De Palma et al., 2021b). To efficiently utilize GPU, we also use batch splits to evaluate multiple subdomains in the same batch as in (Xu et al., 2020; De Palma et al., 2021a). We list our full algorithm $\beta$-CROWN BaB in Appendix D.

**Theorem 3.3.** *$\beta$-CROWN with Branch and Bound on splitting ReLUs is sound and complete.*

*Table 1.* Average runtime and average number of branches on three CIFAR-10 models over 100 properties in the OVAL benchmark.

| Method | CIFAR-10 Base | | | CIFAR-10 Wide | | | CIFAR-10 Deep | | |
|---|---|---|---|---|---|---|---|---|---|
| | time(s) | branches | %timeout | time(s) | branches | %timeout | time(s) | branches | %timeout |
| BaBSR (Bunel et al., 2020b) | 2367.78 | 1020.55 | 36.00 | 2871.14 | 812.65 | 49.00 | 2750.75 | 401.28 | 39.00 |
| MIPplanet (Ehlers, 2017) | 2849.69 | - | 68.00 | 2417.53 | - | 46.00 | 2302.25 | - | 40.00 |
| ERAN* (Singh et al., 2019a; 2018a; 2019b; 2018b) | 805.94 | - | 5.00 | 632.20 | - | 9.00 | 545.72 | - | **0.00** |
| GNN-online (Lu & Kumar, 2020) | 1794.85 | 565.13 | 33.00 | 1367.38 | 372.74 | 15.00 | 1055.33 | 131.85 | 4.00 |
| BDD+ BaBSR (Bunel et al., 2020a) | 807.91 | 195480.14 | 20.00 | 505.65 | 74203.11 | 10.00 | 266.28 | 12722.74 | 4.00 |
| OVAL (BDD+ GNN)* (Bunel et al., 2020a; Lu & Kumar, 2020) | 662.17 | 67938.38 | 16.00 | 280.38 | 17895.94 | 6.00 | 94.69 | 1990.34 | 1.00 |
| A.set BaBSR (De Palma et al., 2021a) | 381.78 | 12004.60 | 7.00 | 165.91 | 2233.10 | 3.00 | 190.28 | 2491.55 | 2.00 |
| BigM+A.set BaBSR (De Palma et al., 2021a) | 390.44 | 11938.75 | 7.00 | 172.65 | 4050.59 | 3.00 | 177.22 | 3275.25 | 2.00 |
| Fast-and-Complete (Xu et al., 2021) | 695.01 | 119522.65 | 17.00 | 495.88 | 80519.85 | 9.00 | 105.64 | 2455.11 | 1.00 |
| BaDNB (BDD+ FSB)* (De Palma et al., 2021b) | 309.29 | 38239.04 | 7.00 | 165.53 | 11214.44 | 4.00 | 10.50 | 368.16 | **0.00** |
| $\beta$-CROWN BaBSR (ours, using the BaBSR branching heuristic) | 233.06 | 276299.50 | 6.00 | 124.26 | 139526.24 | 3.00 | 7.21 | 210.66 | **0.00** |
| $\beta$-CROWN FSB (ours, using the FSB branching heuristic) | **122.71** | 112341.21 | **3.00** | **81.66** | 58565.57 | **2.00** | **6.89** | 42.36 | **0.00** |

\* OVAL (BDD+ GNN) and ERAN are VNN competition 2020 winners; BaDNB (BDD+ FSB) is a concurrent work.

*Figure 1.* Percentage of solved properties with growing running time. $\beta$-CROWN FSB (light green) and $\beta$-CROWN BaBSR (dark green) clearly lead in all 3 settings and solve over 80% properties within 10 seconds.

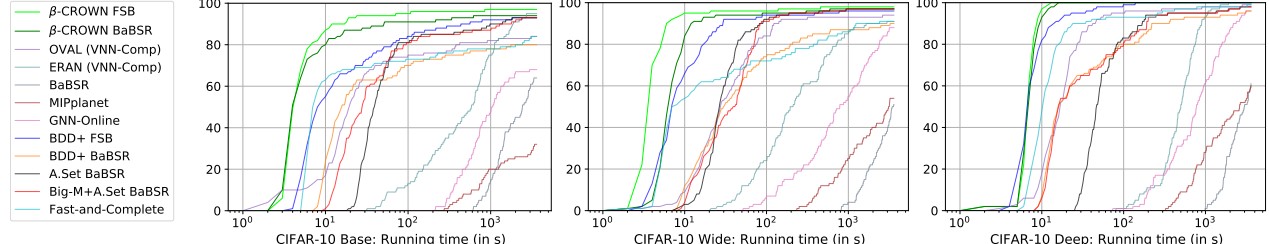

*Table 2.* **Verified accuracy (%)** and avg. per-example verification time (s) on 7 models from SDP-FO (Dathathri et al., 2020). CROWN/DeepPoly are fast but loose bound propagation based methods, and they cannot be improved with more running time. SDP-FO uses stronger semidefinite relaxations, which can be very slow and sometimes has convergence issues. PRIMA is one of the best convex relaxation barrier breaking methods; we did not include kPoly and OptC2V because they are weaker than PRIMA (see Table 4).

| Dataset $\epsilon = 0.3$ and $\epsilon = 2/255$ | Model | CROWN/DeepPoly | | SDP-FO (Dathathri et al., 2020)* | | PRIMA (Müller et al., 2021) | | $\beta$-CROWN FSB | | Upper bound |
|---|---|---|---|---|---|---|---|---|---|---|
| | | Verified% | Time (s) | Ver.% | Time(s) | Ver.% | Time(s) | Ver.% | Time(s) | |
| MNIST | CNN-A-Adv | 1.0 | 1 | 43.4 | >20h | 44.5 | 136 | **68.0** | 76 | 79.5 |
| CIFAR | CNN-B-Adv | 21.5 | 2 | 32.8 | >25h | 38.0 | 360 | **44.5** | 94 | 64.0 |
| | CNN-B-Adv-4 | 43.5 | 2 | 46.0 | >25h | 53.5 | 51 | **54.0** | 52 | 62.5 |
| | CNN-A-Adv | 35.5 | 2 | 39.6 | >25h | 41.5 | 11 | **43.5** | 31 | 52.0 |
| | CNN-A-Adv-4 | 41.5 | 1 | 40.0 | >25h | 45.0 | 7 | **46.0** | 4 | 49.5 |
| | CNN-A-Mix | 23.5 | 1 | 39.6 | >25h | 37.5 | 36 | **41.5** | 33 | 51.5 |
| | CNN-A-Mix-4 | 38.0 | 1 | 47.8 | >25h | 48.5 | 9 | **50.5** | 8 | 55.0 |

\* SDP-FO results are directly from their paper due to very long running time. † PRIMA and $\beta$-CROWN FSB results are on the same set of 200 random examples. $\beta$-CROWN uses **1** GPU and **1** CPU; PRIMA uses **1** GPU and **20** CPUs.

## 4. Experimental Results

**Comparison to Complete Verifiers** We evaluate our method and 10 baselines on complete verification on dataset provided in (Lu & Kumar, 2020; De Palma et al., 2021a) and used in VNN competition 2020 (Liu & Johnson). The dataset contains 3 CIFAR-10 models (Base, Wide, and Deep) with 100 examples each. Each data example is associated with an $\ell_\infty$ norm $\epsilon$ and a target label for verification. The details of each baseline and experimental setups can be found in Appendix E. We report the average verification time and branch numbers in Table 1 and plot the percentage of solved properties over time in Figure 1. $\beta$-CROWN FSB achieves the fastest average running time with minimal timeouts compared to all 10 baselines, and also clearly leads on the cactus plots. Our benefits are more clearly shown in Figure 1, where we solve 80% to 90% examples under 10 seconds and most other verifiers can only verify a small portion or none of the properties within 10 seconds.

**Comparison to Incomplete Verifiers.** In Table 2 we compare against a state-of-the-art semidefinite programming (SDP) based verifier, SDP-FO (Dathathri et al., 2020), and a very recent strong multi-neuron linear relaxation method, PRIMA (Müller et al., 2021), on 1 MNIST and 6 CIFAR-10 models in (Dathathri et al., 2020). The models were trained using adversarial training (verification agnostic), which posed a challenge for verification. The SDP formulation can be tighter than linear relaxation based ones, but it takes 2 to 3 hours to converge on one GPU for verifying a single property, resulting 5,400 GPU hours to verify 200 testing images with 10 labels. Due to resource limitations, we directly quote SDP-FO results from (Dathathri et al., 2020) on the same set of models, and evaluate verified accuracy on the same set of 200 test images for all other baselines. Table 2 shows that overall we are three orders of magnitude faster than SDP-FO while still achieving consistently higher verified accuracy. Additional results are in Appendix E.3.

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

# A. Additional Background

## A.1. The NN Verification Problem and the LP Verifier.

Neural network verification seeks the solution of the optimization problem in Eq. 5:

$$\min f(x) := z^{(L)}(x) \quad \text{s.t. } z^{(i)} = \mathbf{W}^{(i)}\hat{z}^{(i-1)} + \mathbf{b}^{(i)}, \hat{z}^{(i)} = \sigma(z^{(i)}), x \in \mathcal{C}, i \in \{1, \cdots, L-1\} \tag{5}$$

The set $\mathcal{C}$ defines the allowed input region and our aim is to find the minimum of $f(x)$ for $x \in \mathcal{C}$, and throughout this paper we consider $\mathcal{C}$ as an $\ell_\infty$ ball around a data example $x_0$: $\mathcal{C} = \{x \mid \|x - x_0\|_\infty \leq \epsilon\}$ but other $\ell_p$ norms can also be supported. In practical settings, we typically have "specifications" to verify, which are (usually linear) functions of neural network outputs describing the desired behavior of neural networks. For example, to guarantee robustness we typically investigate the margin between logits. Because the specification can also be seen as an output layer of NN and merged into $f(x)$ under verification, we do not discuss it in detail in this work. We consider the canonical specification $f(x) > 0$: if we can prove that $f(x) > 0$, $\forall x \in \mathcal{C}$, we say $f(x)$ is verified.

A commonly used incomplete verification technique is to relax non-convex ReLUs with linear constraints and turn the verification problem into a linear programming (LP) problem, which can then be solved with linear solvers. We refer to it as "LP verifier" in this paper. Specifically, given $\text{ReLU}(z_j^{(i)}) := \max(0, z_j^{(i)})$ and its intermediate layer bounds $\mathbf{l}_j^{(i)} \leq z_j^{(i)} \leq \mathbf{u}_j^{(i)}$, each ReLU can be categorized into three cases linearly relaxed differently: (1) if $\mathbf{l}_j^{(i)} \geq 0$ (ReLU in linear region) then $\hat{z}_j^{(i)} = z_j^{(i)}$; (2) if $\mathbf{u}_j^{(i)} \leq 0$ (ReLU in inactive region) then $\hat{z}_j^{(i)} = 0$; (3) if $\mathbf{l}_j^{(i)} \leq 0 \leq \mathbf{u}_j^{(i)}$ (ReLU is *unstable*) then three linear bounds are used: $\hat{z}_j^{(i)} \geq 0$, $\hat{z}_j^{(i)} \geq z_j^{(i)}$, and $\hat{z}_j^{(i)} \leq \frac{\mathbf{u}_j^{(i)}}{\mathbf{u}_j^{(i)} - \mathbf{l}_j^{(i)}}\left(z_j^{(i)} - \mathbf{l}_j^{(i)}\right)$; they are often referred to as the "triangle" relaxation (Ehlers, 2017; Wong & Kolter, 2018). The intermediate layer bounds $\mathbf{l}^{(i)}$ and $\mathbf{u}^{(i)}$ are usually obtained from a cheaper bound propagation method (see next subsection). LP verifiers can provide relatively tight bounds but linear solvers are still expensive especially when the network is large. Also, unlike our $\beta$-CROWN, they have to use fixed intermediate bounds and cannot use the joint optimization of intermediate layer bounds (Section B.3) to tighten relaxation.

## A.2. CROWN: Efficient Incomplete Verification by Propagating Linear Bounds

Another cheaper way to give a lower bound for the verification problem Eq. 5 is through sound bound propagation. CROWN (Zhang et al., 2018) is a representative method that propagates a linear bound of $f(x)$ w.r.t. every intermediate layer in a backward manner until reaching the input $x$. CROWN uses two linear constraints to relax unstable ReLU neurons: a linear upper bound $\hat{z}_j^{(i)} \leq \frac{\mathbf{u}_j^{(i)}}{\mathbf{u}_j^{(i)} - \mathbf{l}_j^{(i)}}\left(z_j^{(i)} - \mathbf{l}_j^{(i)}\right)$ and a linear lower bound $\hat{z}_j^{(i)} \geq \boldsymbol{\alpha}_j^{(i)} z_j^{(i)}$ $(0 \leq \boldsymbol{\alpha}_j^{(i)} \leq 1)$. We can then bound the output of a ReLU layer:

**Lemma A.1** (ReLU relaxation in CROWN). *Given $w, v \in \mathbb{R}^d, \mathbf{u} \leq v \leq \mathbf{l}$ (element-wise), we have*

$$w^\top \text{ReLU}(v) \geq w^\top \mathbf{D}v + b',$$

*where $\mathbf{D}$ is a diagonal matrix containing free variables $0 \leq \boldsymbol{\alpha}_j \leq 1$ only when $\mathbf{u}_j > 0 > \mathbf{l}_j$ and $w_j \geq 0$, while its rest values as well as constant $b'$ are determined by $\mathbf{l}, \mathbf{u}, w$.*

Detailed forms of each term are listed in Appendix C. Lemma A.1 can be repeatedly applied, resulting in an efficient back-substitution procedure to derive a linear lower bound of NN output w.r.t. $x$:

**Lemma A.2** (CROWN bound (Zhang et al., 2018)). *Given an L-layer ReLU NN $f(x) : \mathbb{R}^{d_0} \to \mathbb{R}$ with weights $\mathbf{W}^{(i)}$, biases $\mathbf{b}^{(i)}$, pre-ReLU bounds $\mathbf{l}^{(i)} \leq z^{(i)} \leq \mathbf{u}^{(i)}$ $(1 \leq i \leq L)$ and input constraint $x \in \mathcal{C}$. We have*

$$\min_{x \in \mathcal{C}} f(x) \geq \min_{x \in \mathcal{C}} \boldsymbol{a}_{\text{CROWN}}^\top x + c_{\text{CROWN}}$$

*where $\boldsymbol{a}_{\text{CROWN}}$ and $c_{\text{CROWN}}$ can be computed using $\mathbf{W}^{(i)}, \mathbf{b}^{(i)}, \mathbf{l}^{(i)}, \mathbf{u}^{(i)}$ in polynomial time.*

When $\mathcal{C}$ is an $\ell_p$ norm ball, minimization over the linear function can be easily solved using Hölder's inequality. The main benefit of CROWN is its efficiency: CROWN can be efficiently implemented on machine learning accelerators such as GPUs (Xu et al., 2020) and TPUs (Zhang et al., 2020), and it can be a few magnitudes faster than an LP verifier which is hard to parallelize on GPUs. CROWN was generalized to general architectures (Xu et al., 2020; Shi et al., 2020) while we only demonstrate it for feedforward ReLU networks for simplicity.

### A.3. Branch and Bound and Neuron Split Constraints

Branch and bound (BaB) method is widely adopted in complete verifiers (Bunel et al., 2018): we divide the domain of the verification problem $\mathcal{C}$ into two subdomains $\mathcal{C}_1 = \{x \in \mathcal{C}, z_j^{(i)} \geq 0\}$ and $\mathcal{C}_2 = \{x \in \mathcal{C}, z_j^{(i)} < 0\}$ where $z_j^{(i)}$ is an unstable ReLU neuron in $\mathcal{C}$ but now becomes linear for each subdomain. Incomplete verifiers can then estimate the lower bound of each subdomain with relaxations. If the lower bound produced for subdomain $\mathcal{C}_i$ (denoted by $\underline{f}_{\mathcal{C}_i}$) is greater than 0, $\mathcal{C}_i$ is verified; otherwise, we further branch over domain $\mathcal{C}_i$ by splitting another unstable ReLU neuron. The process terminates when all subdomains are verified. The completeness is guaranteed when all unstable ReLU neurons are split.

**LP verifier with neuron split constraints.** A popular incomplete verifier used in BaB is LP verifier. Essentially, when we split the $j$-th ReLU in layer $i$, we can simply add $z_j^{(i)} \geq 0$ or $z_j^{(i)} < 0$ to Eq. 5 and get a linearly relaxed lower bound to each subdomain. We denote the $\mathcal{Z}^{+(i)}$ and $\mathcal{Z}^{-(i)}$ as the set of neuron indices with positive and negative split constraints in layer $i$. We define the split constraints at layer $i$ as $\mathcal{Z}^{(i)} := \{z^{(i)} \mid z_{j_1}^{(i)} \geq 0, z_{j_2}^{(i)} < 0, \forall j_1 \in \mathcal{Z}^{+(i)}, \forall j_2 \in \mathcal{Z}^{-(i)}\}$. We denote the vector of all pre-ReLU neurons as $z$, and we define a set $\mathcal{Z}$ to represent the split constraints on $z$: $\mathcal{Z} = \mathcal{Z}^{(1)} \cap \mathcal{Z}^{(2)} \cap \cdots \cap \mathcal{Z}^{(L-1)}$. For convenience, we also use the shorthand $\tilde{\mathcal{Z}}^{(i)} := \mathcal{Z}^{(1)} \cap \cdots \cap \mathcal{Z}^{(i)}$ and $\tilde{z}^{(i)} := \{z^{(1)}, z^{(2)}, \cdots, z^{(i)}\}$. LP verifiers can easily handle these neuron split constraints but are very expensive than bound propagation methods like CROWN and cannot be accelerated on GPUs.

**Branching strategy.** Branching strategies (selecting which ReLU neuron to split) are generally agnostic to the incomplete verifier used in BaB but do affect the overall BaB performance. BaBSR (Bunel et al., 2020b) is a widely used strategy in complete verifiers, which is based on an fast estimates on objective improvements after splitting each neuron. The neuron with highest estimated improvement is selected for branching. Recently, Filtered Smart Branching (FSB) (De Palma et al., 2021b) improves BaBSR by mimicking strong branching - it utilizes bound propagation methods to evaluate the best a few candidates proposed by BaBSR and chooses the one with largest improvement. Graph neural network (GNN) based branching was also proposed (Lu & Kumar, 2020). Our $\beta$-CROWN BaB is a general complete verification framework fit for any potential branching strategy, and we evaluate both BaBSR and FSB in experiments.

## B. Additional Technical Details

In main text we have given a brief overview of the our $\beta$-CROWN algorithm. In this section, we will go over the details of the bound propagation processes with split constraints, derive the $\beta$-CROWN bounds from both primal space and dual space, and discuss how to tighten the bound using free parameters $\boldsymbol{\alpha}$ and $\boldsymbol{\beta}$. Lastly, we describe $\beta$-CROWN BaB, a complete verifier that also becomes a strong incomplete verifier when stopped early.

### B.1. $\beta$-CROWN: Linear Bound Propagation with Neuron Split Constraints

The NN verification problem under neuron split constraints can be written as an optimization:

$$\min_{x \in \mathcal{C}, z \in \mathcal{Z}} f(x). \tag{6}$$

Bound propagation methods like CROWN can give a relatively tight lower bound for $\min_{x \in \mathcal{C}} f(x)$ but they *cannot handle the neuron split constraints* $z \in \mathcal{Z}$. Before we present our main theorem, we first show the intuition on how to apply split constraints to the bound propagation process.

To encode the neuron splits, we first define diagonal matrix $\mathbf{S}^{(i)} \in \mathbb{R}^{d_i \times d_i}$ in Eq. 7 where $i \in [1, \cdots L-1], j \in [1, \cdots, d_i]$:

$$\mathbf{S}_{j,j}^{(i)} = -1 (\text{if split } z_j^{(i)} \geq 0); \quad \mathbf{S}_{j,j}^{(i)} = +1 (\text{if split } z_j^{(i)} < 0); \quad \mathbf{S}_{j,j}^{(i)} = 0 (\text{if not split } z_j^{(i)}) \tag{7}$$

We start from last layer and derive linear bounds for each intermediate layer $z^{(i)}$ and $\hat{z}^{(i)}$ for both $x \in \mathcal{C}$ and $z \in \mathcal{Z}$. We also assume that pre-ReLU bounds $\mathbf{l}^{(i)} \leq z^{(i)} \leq \mathbf{u}^{(i)}$ for each layer $i$ are available. We initially have:

$$\min_{x \in \mathcal{C}, z \in \mathcal{Z}} f(x) = \min_{x \in \mathcal{C}, z \in \mathcal{Z}} \mathbf{W}^{(L)} \hat{z}^{(L-1)} + \mathbf{b}^{(L)}. \tag{8}$$

Since $\hat{z}^{(L-1)} = \text{ReLU}(z^{(L-1)})$, we can apply Lemma A.1 to relax the ReLU neuron at layer $L-1$, and obtain a linear lower bound for $f(x)$ w.r.t. $z^{(L-1)}$:

$$\min_{x \in \mathcal{C}, z \in \mathcal{Z}} f(x) \geq \min_{x \in \mathcal{C}, z \in \mathcal{Z}} \mathbf{W}^{(L)} \mathbf{D}^{(L-1)} z^{(L-1)} + \text{const.}$$

To enforce the split neurons at layer $L-1$, we use a Lagrange function with $\boldsymbol{\beta}^{(L-1)\top} \mathbf{S}^{(L-1)}$ multiplied on $z^{(L-1)}$:

$$\min_{x \in \mathcal{C}, z \in \mathcal{Z}} f(x) \geq \min_{\substack{x \in \mathcal{C} \\ \hat{z}^{(L-2)} \in \hat{\mathcal{Z}}^{(L-2)}}} \max_{\boldsymbol{\beta}^{(L-1)} \geq 0} \mathbf{W}^{(L)} \mathbf{D}^{(L-1)} z^{(L-1)} + \boldsymbol{\beta}^{(L-1)\top} \mathbf{S}^{(L-1)} z^{(L-1)} + \text{const}$$

$$\geq \max_{\boldsymbol{\beta}^{(L-1)} \geq 0} \min_{\substack{x \in \mathcal{C} \\ \hat{z}^{(L-2)} \in \hat{\mathcal{Z}}^{(L-2)}}} \left( \mathbf{W}^{(L)} \mathbf{D}^{(L-1)} + \boldsymbol{\beta}^{(L-1)\top} \mathbf{S}^{(L-1)} \right) z^{(L-1)} + \text{const} \tag{9}$$

The first inequality is due to the definition of the Lagrange function: we remove the constraint $z^{(L-1)} \in \mathcal{Z}^{(L-1)}$ and use a multiplier to replace this constraint. The second inequality is due to weak duality. Due to the design of $\mathbf{S}^{(L-1)}$, neuron split $z_j^{(L-1)} \geq 0$ has a negative multiplier $-\boldsymbol{\beta}_j^{(L-1)}$ and split $z_j^{(L-1)} < 0$ has a positive multiplier $\boldsymbol{\beta}_j^{(L-1)}$. Any $\boldsymbol{\beta}^{(L-1)} \geq 0$ yields a lower bound for the constrained optimization. Then we substitute $z^{(L-1)}$ with $\mathbf{W}^{(L-1)} \hat{z}^{(L-2)} + \mathbf{b}^{(L-1)}$ for next layer:

$$\min_{x \in \mathcal{C}, z \in \mathcal{Z}} f(x) \geq \max_{\boldsymbol{\beta}^{(L-1)} \geq 0} \min_{\substack{x \in \mathcal{C} \\ \hat{z}^{(L-2)} \in \hat{\mathcal{Z}}^{(L-2)}}} \left( \mathbf{W}^{(L)} \mathbf{D}^{(L-1)} + \boldsymbol{\beta}^{(L-1)\top} \mathbf{S}^{(L-1)} \right) \mathbf{W}^{(L-1)} \hat{z}^{(L-2)} + \text{const} \tag{10}$$

We define a matrix $\mathbf{A}^{(i)}$ to represent the linear relationship between $f(x)$ and $\hat{z}^{(i)}$, where $\mathbf{A}^{(L-1)} = \mathbf{W}^{(L)}$ according to Eq. 8 and $\mathbf{A}^{(L-2)} = (\mathbf{A}^{(L-1)} \mathbf{D}^{(L-1)} + \boldsymbol{\beta}^{(L-1)\top} \mathbf{S}^{(L-1)}) \mathbf{W}^{(L-1)}$ by Eq. 10. Considering 1-dimension output $f(x)$, $\mathbf{A}^{(i)}$ has only 1 row. With $\mathbf{A}^{(L-2)}$, Eq. 10 becomes:

$$\min_{x \in \mathcal{C}, z \in \mathcal{Z}} f(x) \geq \max_{\boldsymbol{\beta}^{(L-1)} \geq 0} \min_{\substack{x \in \mathcal{C} \\ \hat{z}^{(L-2)} \in \hat{\mathcal{Z}}^{(L-2)}}} \mathbf{A}^{(L-2)} \hat{z}^{(L-2)} + \text{const},$$

which is in a form similar to Eq. 8 except for the outer maximization over $\boldsymbol{\beta}^{(L-1)}$. This allows the back-substitution process (Eq. 8, Eq. 9, and Eq. 10) to continue. In each step, we swap $\max$ and $\min$ as in Eq. 9, so every maximization over $\boldsymbol{\beta}^{(i)}$ is outside of $\min_{x \in \mathcal{C}}$. Eventually, we have:

$$\min_{x \in \mathcal{C}, z \in \mathcal{Z}} f(x) \geq \max_{\boldsymbol{\beta} \geq 0} \min_{x \in \mathcal{C}} \mathbf{A}^{(0)} x + \text{const},$$

where $\boldsymbol{\beta} := \left[ \boldsymbol{\beta}^{(1)\top} \ \boldsymbol{\beta}^{(2)\top} \ \cdots \ \boldsymbol{\beta}^{(L-1)\top} \right]^{\top}$ concatenates all $\boldsymbol{\beta}^{(i)}$ vectors. Following the above idea, we present the main theorem in Theorem B.1 (proof is given in Appendix C).

**Theorem B.1** ($\beta$-CROWN bound). *Given an $L$-layer NN $f(x) : \mathbb{R}^{d_0} \to \mathbb{R}$ with weights $\mathbf{W}^{(i)}$, biases $\mathbf{b}^{(i)}$, pre-ReLU bounds $\mathbf{l}^{(i)} \leq z^{(i)} \leq \mathbf{u}^{(i)}$ ($1 \leq i \leq L$), input bounds $\mathcal{C}$, split constraints $\mathcal{Z}$. We have:*

$$\min_{x \in \mathcal{C}, z \in \mathcal{Z}} f(x) \geq \max_{\boldsymbol{\beta} \geq 0} \min_{x \in \mathcal{C}} (\boldsymbol{a} + \mathbf{P}\boldsymbol{\beta})^{\top} x + \mathbf{q}^{\top}\boldsymbol{\beta} + c, \tag{11}$$

*where $\boldsymbol{a} \in \mathbb{R}^{d_0}, \mathbf{P} \in \mathbb{R}^{d_0 \times (\sum_{i=1}^{L-1} d_i)}, \mathbf{q} \in \mathbb{R}^{\sum_{i=1}^{L-1} d_i}$ and $c \in \mathbb{R}$ are functions of $\mathbf{W}^{(i)}$, $\mathbf{b}^{(i)}$, $\mathbf{l}^{(i)}$, $\mathbf{u}^{(i)}$.*

Detailed formulations for $\boldsymbol{a}$, $\mathbf{P}$, $\mathbf{q}$ and $c$ are given in Appendix C. Theorem B.1 shows that when neuron split constraints exist, $f(x)$ can still be bounded by a linear equation containing optimizable multipliers $\boldsymbol{\beta}$. Observing Eq. 9, the main difference between CROWN and $\beta$-CROWN lies in the relaxation of each ReLU layer, where we need an extra term $\boldsymbol{\beta}^{(i)\top} \mathbf{S}^{(i)}$ in the linear relationship matrix (for example, $\mathbf{W}^{(L)} \mathbf{D}^{(L-1)}$ in Eq. 9) between $f(x)$ and $z^{(i)}$ to enforce neuron split constraints. This extra term in every ReLU layer yields $\mathbf{P}$ and $\mathbf{q}$ in Eq. 11 after bound propagations.

To solve the optimization problem in Eq. 11, we note that in the $\ell_p$ norm robustness setting ($\mathcal{C} = \{x \mid \|x - x_0\|_p \leq \epsilon\}$), the inner minimization has a closed solution:

$$\min_{x \in \mathcal{C}, z \in \mathcal{Z}} f(x) \geq \max_{\boldsymbol{\beta} \geq 0} -\|\boldsymbol{a} + \mathbf{P}\boldsymbol{\beta}\|_q \epsilon + (\mathbf{P}^{\top} x_0 + \mathbf{q})^{\top}\boldsymbol{\beta} + \boldsymbol{a}^{\top} x_0 + c := \max_{\boldsymbol{\beta} \geq 0} g(\boldsymbol{\beta}) \tag{12}$$

where $\frac{1}{p} + \frac{1}{q} = 1$. The maximization is concave in $\boldsymbol{\beta}$ ($q \geq 1$), so we can simply optimize it using projected (super)gradient ascent with gradients from an automatic differentiation library. Since any $\boldsymbol{\beta} \geq 0$ yields a valid lower bound for $\min_{x \in \mathcal{C}, z \in \mathcal{Z}} f(x)$, convergence is not necessary to guarantee soundness. $\beta$-CROWN is efficient - it has the same asymptotic

complexity as CROWN when $\boldsymbol{\beta}$ is fixed. When $\boldsymbol{\beta} = 0$, $\beta$-CROWN yields the same results as CROWN; however the additional optimizable $\boldsymbol{\beta}$ allows us to maximize and tighten the lower bound due to neuron split constraints.

We define $\boldsymbol{\alpha}^{(i)} \in \mathbb{R}^{d_i}$ for free variables associated with unstable ReLU neurons in Lemma A.1 for layer $i$ and define all free variables $\boldsymbol{\alpha} = \{\boldsymbol{\alpha}^{(1)} \cdots \boldsymbol{\alpha}^{(L-1)}\}$. Since any $0 \leq \boldsymbol{\alpha}_j^{(i)} \leq 1$ yields a valid bound, we can optimize it to tighten the bound. Formally, we rewrite Eq. 12 with $\boldsymbol{\alpha}$ explicitly:

$$\min_{x \in \mathcal{C}, z \in \mathcal{Z}} f(x) \geq \max_{0 \leq \boldsymbol{\alpha} \leq 1, \, \boldsymbol{\beta} \geq 0} g(\boldsymbol{\alpha}, \boldsymbol{\beta}). \tag{13}$$

### B.2. Connections to the Dual Problem

In this subsection, we show that $\beta$-CROWN can also be derived from a dual LP problem. Based on Eq. 5 and linear relaxations in Section A.1, we first construct an LP problem for $\ell_\infty$ robustness verification in Eq. 14 where $i \in \{1, \cdots, L-1\}$.

$$\min \ f(x) := z^{(L)}(x) \quad \text{s.t.}$$

Network and Input Bounds: $z^{(i)} = \mathbf{W}^{(i)} \hat{z}^{(i-1)} + \mathbf{b}^{(i)}; \hat{z}^{(0)} \geq x_0 - \epsilon; \hat{z}^{(0)} \leq x_0 + \epsilon;$

Stable: $\hat{z}_j^{(i)} = z_j^{(i)}$ (if $\mathbf{l}_j^{(i)} \geq 0$); $\hat{z}_j^{(i)} = 0$ (if $\mathbf{u}_j^{(i)} \leq 0$); $\tag{14}$

Unstable: $\hat{z}_j^{(i)} \geq 0, \hat{z}_j^{(i)} \geq z_j^{(i)}, \hat{z}_j^{(i)} \leq \frac{\mathbf{u}_j^{(i)}}{\mathbf{u}_j^{(i)} - \mathbf{l}_j^{(i)}} \left( z_j^{(i)} - \mathbf{l}_j^{(i)} \right)$ (if $\mathbf{l}_j^{(i)} < 0 < \mathbf{u}_j^{(i)}, j \notin \mathcal{Z}^{+(i)} \cup \mathcal{Z}^{-(i)}$)

Neuron Split Constraints: $\hat{z}_j^{(i)} = z_j^{(i)}, z_j^{(i)} \geq 0$ (if $j \in \mathcal{Z}^{+(i)}$); $\hat{z}_j^{(i)} = 0, z_j^{(i)} < 0$ (if $j \in \mathcal{Z}^{-(i)}$)

Compared to the formulation in (Wong & Kolter, 2018), we have neuron split constraints. Many BaB based complete verifiers (Bunel et al., 2018; Lu & Kumar, 2020) use an LP solver for Eq. 14 as the incomplete verifier. We first show that it is possible to derive Theorem B.1 from the dual of this LP, leading to Theorem B.2:

**Theorem B.2.** *The objective $d_{LP}$ for the dual problem of Eq. 14 can be represented as*

$$d_{LP} = -\|\mathbf{a} + \mathbf{P}\boldsymbol{\beta}\|_1 \cdot \epsilon + (\mathbf{P}^\top x_0 + \mathbf{q})^\top \boldsymbol{\beta} + \mathbf{a}^\top x_0 + c,$$

*where $\mathbf{a}$, $\mathbf{P}$, $\mathbf{q}$ and $c$ are defined in the same way as in Theorem B.1, and $\boldsymbol{\beta} \geq 0$ corresponds to the dual variables of neuron split constraints in Eq. 14.*

A similar connection between CROWN and dual LP based verifier (Wong & Kolter, 2018) was shown in (Salman et al., 2019), and their results can be seen as a special case of ours when $\boldsymbol{\beta} = 0$ (none of the split constraints are active). An immediate consequence is that $\beta$-CROWN can potentially solve Eq. 14 as well as using an LP solver:

**Corollary B.2.1.** *When $\boldsymbol{\alpha}$ and $\boldsymbol{\beta}$ are optimally set and intermediate bounds $\mathbf{l}, \mathbf{u}$ are fixed, $\beta$-CROWN produces $p_{LP}^*$, the optimal objective of LP with split constraints in Eq. 14:*

$$\max_{0 \leq \boldsymbol{\alpha} \leq 1, \boldsymbol{\beta} \geq 0} g(\boldsymbol{\alpha}, \boldsymbol{\beta}) = p_{LP}^*,$$

In Appendix C, we give detailed formulations for conversions between the variables $\boldsymbol{\alpha}$, $\boldsymbol{\beta}$ in $\beta$-CROWN and their corresponding dual variables in the LP problem.

### B.3. Joint Optimization of Free Variables in $\beta$-CROWN

In Eq. 13, $g$ is also a function of $\mathbf{l}_j^{(i)}$ and $\mathbf{u}_j^{(i)}$, the intermediate layer bounds for each neuron $z_j^{(i)}$. They are also computed using $\beta$-CROWN. To obtain $\mathbf{l}_j^{(i)}$, we set $f(x) := z_j^{(i)}(x)$ and apply Theorem B.1:

$$\min_{x \in \mathcal{C}, \tilde{z}^{(i-1)} \in \tilde{\mathcal{Z}}^{(i-1)}} z_j^{(i)}(x) \geq \max_{0 \leq \boldsymbol{\alpha}' \leq 1, \, \boldsymbol{\beta}' \geq 0} g'(\boldsymbol{\alpha}', \boldsymbol{\beta}') := \mathbf{l}_j^{(i)} \tag{15}$$

and for $\mathbf{u}_j^{(i)}$ we set $f(x) := -z_j^{(i)}(x)$. Importantly, during solving these intermediate layer bounds, the $\boldsymbol{\alpha}'$ and $\boldsymbol{\beta}'$ are *independent sets of variables*, not the same ones for the objective $f(x) := z^{(L)}$. Since $g$ is a function of $\mathbf{l}_j^{(i)}$, it is also a function of $\boldsymbol{\alpha}'$ and $\boldsymbol{\beta}'$. In fact, there are a total of $\sum_{i=1}^{L-1} d_i$ intermediate layer neurons, and each neuron is associated with a set of independent $\boldsymbol{\alpha}'$ and $\boldsymbol{\beta}'$ variables. Optimizing these variables allowing us to tighten the relaxations on unstable ReLU

neurons (which depend on $\mathbf{l}_j^{(i)}$ and $\mathbf{u}_j^{(i)}$) and produce tight final bounds, which is impossible in LP. In other words, we need to optimize $\hat{\alpha}$ and $\hat{\beta}$, which are two vectors concatenating $\alpha, \beta$ as well as a large number of $\alpha'$ and $\beta'$ used to compute each intermediate layer bound:

$$\min_{x \in \mathcal{C}, z \in \mathcal{Z}} f(x) \geq \max_{0 \leq \hat{\alpha} \leq 1, \, \hat{\beta} \geq 0} g(\hat{\alpha}, \hat{\beta}). \tag{16}$$

This formulation is non-convex and has a large number of variables. Since any $0 \leq \hat{\alpha} \leq 1, \hat{\beta} \geq 0$ leads to a valid lower bound, the non-convexity does not affect soundness. When intermediate layer bounds are also allowed to be tightened during optimization, we can outperform the LP verifier for Eq. 14 using fixed intermediate layer bounds. Typically, when Eq. 14 is formed, intermediate layer bounds are pre-computed with bound propagation procedures (Bunel et al., 2018; Lu & Kumar, 2020), which are far from optimal.

To estimate the dimension of this problem, we denote the number of unstable neurons at layer $i$ as $s_i := \text{Tr}(|\mathbf{S}^{(i)}|)$. Each neuron in layer $i$ is associated with $2 \times \sum_{k=1}^{i-1} s_k$ variables $\alpha'$. Suppose each hidden layer has $d$ neurons ($s_i = O(d)$), then $\hat{\alpha}$ has $2 \times \sum_{i=1}^{L-1} d_i \sum_{k=1}^{i-1} s_k = O(L^2 d^2)$ variables in total. This can be too large for efficient optimization, so we share $\alpha'$ and $\beta'$ among the intermediate neurons of the same layer, leading to a total number of $O(L^2 d)$ variables to optimize. Note that a weaker form of joint optimization was also discussed in (Xu et al., 2021) without $\beta$, and a detailed analysis can be found in Appendix D.2.

### B.4. $\beta$-CROWN with Branch and Bound ($\beta$-CROWN BaB)

We perform complete verification following BaB framework (Bunel et al., 2018) using $\beta$-CROWN as the incomplete solver, and we use simple branching heuristics like BaBSR (Bunel et al., 2020b) or FSB (De Palma et al., 2021b). To efficiently utilize GPU, we also use batch splits to evaluate multiple subdomains in the same batch as in (Xu et al., 2020; De Palma et al., 2021a). We list our full algorithm $\beta$-CROWN BaB in Appendix D and we show it is sound and complete here:

**Theorem B.3.** *$\beta$-CROWN with Branch and Bound on splitting ReLUs is sound and complete.*

Soundness is trivial because $\beta$-CROWN is a sound verifier. For completeness, it suffices to show that when all unstable ReLU neurons are split, $\beta$-CROWN gives the global minimum for Eq. 14. In contrast, combining CROWN (Zhang et al., 2018) with BaB does *not* yield a complete verifier, as it cannot detect infeasible splits and a slow LP solver is still needed to guarantee completeness (Xu et al., 2021). Instead, $\beta$-CROWN can detect infeasible subdomains - according to duality theory, an infeasible primal problem leads to an unbounded dual objective, which can be detected (see Sec. D.3 for more details).

Additionally, we show the potential of *early stopping a complete verifier as an incomplete verifier*. BaB approaches the exact solution of Eq. 5 by splitting the problem into multiple subdomains, and more subdomains give a tighter lower bound for Eq. 5. Unlike traditional complete verifiers, $\beta$-CROWN is efficient to explore a large number of subdomains during a very short time, making $\beta$-CROWN BaB an attractive solution for efficient incomplete verification.

## C. Proofs for $\beta$-CROWN

### C.1. Proofs for deriving $\beta$-CROWN using bound propagation

Lemma A.1 is from part of the proof of the main theorem in Zhang et al. (2018). Here we present it separately to use it as an useful subprocedure for our later proofs.

**Lemma A.1** (Relaxation of a ReLU layer in CROWN). *Given two vectors $w, v \in \mathbb{R}^d, \mathbf{u} \leq v \leq \mathbf{l}$ (element-wise), we have*

$$w^\top \text{ReLU}(v) \geq w^\top \mathbf{D} v + b',$$

*where $\mathbf{D}$ is a diagonal matrix defined as:*

$$\mathbf{D}_{j,j} = \begin{cases} 1, & \text{if } \mathbf{l}_j \geq 0 \\ 0, & \text{if } \mathbf{u}_j \leq 0 \\ \alpha_j, & \text{if } \mathbf{u}_j > 0 > \mathbf{l}_j \text{ and } w_j \geq 0 \\ \frac{\mathbf{u}_j}{\mathbf{u}_j - \mathbf{l}_j}, & \text{if } \mathbf{u}_j > 0 > \mathbf{l}_j \text{ and } w_j < 0, \end{cases} \tag{17}$$

$0 \leq \boldsymbol{\alpha}_j \leq 1$ *are free variables,* $b' = w^\top \underline{\mathbf{b}}$ *and each element in* $\underline{\mathbf{b}}$ *is*

$$
\underline{\mathbf{b}}_j = \begin{cases} 0, & \text{if } \mathbf{l}_j > 0 \text{ or } \mathbf{u}_j \leq 0 \\ 0, & \text{if } \mathbf{u}_j > 0 > \mathbf{l}_j \text{ and } w_j \geq 0 \\ -\frac{\mathbf{u}_j \mathbf{l}_j}{\mathbf{u}_j - \mathbf{l}_j}, & \text{if } \mathbf{u}_j > 0 > \mathbf{l}_j \text{ and } w_j < 0. \end{cases} \tag{18}
$$

*Proof.* For the $j$-th ReLU neuron, if $\mathbf{l}_j \geq 0$, then $\mathrm{ReLU}(v_j) = v_j$; if $\mathbf{u}_j < 0$, then $\mathrm{ReLU}(v_j) = 0$. For the case of $\mathbf{l}_j < 0 < \mathbf{u}_j$, the ReLU function can be linearly upper and lower bounded within this range:

$$
\boldsymbol{\alpha}_j v_j \leq \mathrm{ReLU}(v_j) \leq \frac{\mathbf{u}_j}{\mathbf{u}_j - \mathbf{l}_j} (v_j - \mathbf{l}_j) \quad \forall \mathbf{l}_j \leq v_j \leq \mathbf{u}_j
$$

where $0 \leq \boldsymbol{\alpha}_j \leq 1$ is a free variable - any value between 0 and 1 produces a valid lower bound. To lower bound $w^\top \mathrm{ReLU}(v) = \sum_j w_j \mathrm{ReLU}(v_j)$, for each term in this summation, we take the lower bound of $\mathrm{ReLU}(v_j)$ if $w_j$ is positive and take the upper bound of $\mathrm{ReLU}(v_j)$ if $w_j$ is negative (reflected in the definitions of $\mathbf{D}$ and $\underline{\mathbf{b}}$). This conservative choice allows us to always obtain a lower bound $\forall \mathbf{l} \leq v \leq \mathbf{u}$:

$$
\sum_j w_j \mathrm{ReLU}(v_j) \geq \sum_j w_j \left( \mathbf{D}_{j,j} v_j + \underline{\mathbf{b}}_j \right) = w^\top \mathbf{D} v + w^\top \underline{\mathbf{b}} = w^\top \mathbf{D} v + b'
$$

where $\mathbf{D}_{j,j}$ and $\underline{\mathbf{b}}_j$ are defined in Eq. 17 and Eq. 18 representing the lower or upper bounds of ReLU. $\qquad\square$

Before proving our main theorem (Theorem B.1), we first define matrix $\boldsymbol{\Omega}$, which is the product of a series of model weights $\mathbf{W}$ and "weights" for relaxed ReLU layers $\mathbf{D}$:

**Definition C.1.** *Given a set of matrices* $\mathbf{W}^{(2)}, \cdots, \mathbf{W}^{(L)}$ *and* $\mathbf{D}^{(1)}, \cdots, \mathbf{D}^{(L-1)}$, *we define a recursive function* $\boldsymbol{\Omega}(k, i)$ *for* $1 \leq i \leq k \leq L$ *as*

$$
\boldsymbol{\Omega}(i, i) = \mathbf{I}, \ \boldsymbol{\Omega}(k+1, i) = \mathbf{W}^{(k+1)} \mathbf{D}^{(k)} \boldsymbol{\Omega}(k, i)
$$

For example, $\boldsymbol{\Omega}(3, 1) = \mathbf{W}^{(3)} \mathbf{D}^{(2)} \mathbf{W}^{(2)} \mathbf{D}^{(1)}$, $\boldsymbol{\Omega}(5, 2) = \mathbf{W}^{(5)} \mathbf{D}^{(4)} \mathbf{W}^{(4)} \mathbf{D}^{(3)} \mathbf{W}^{(3)} \mathbf{D}^{(2)}$. Now we present our main theorem with each term explicitly written:

**Theorem B.1** ($\beta$-CROWN bound). *Given a L-layer neural network* $f(x) : \mathbb{R}^{d_0} \to \mathbb{R}$ *with weights* $\mathbf{W}^{(i)}$, *biases* $\mathbf{b}^{(i)}$, *pre-ReLU bounds* $\mathbf{l}^{(i)} \leq z^{(i)} \leq \mathbf{u}^{(i)}$ ($1 \leq i \leq L$), *input constraint* $\mathcal{C}$ *and split constraint* $\mathcal{Z}$. *We have*

$$
\min_{x \in \mathcal{C}, z \in \mathcal{Z}} f(x) \geq \max_{\boldsymbol{\beta} \geq 0} \min_{x \in \mathcal{C}} (\boldsymbol{a} + \mathbf{P}\boldsymbol{\beta})^\top x + \mathbf{q}^\top \boldsymbol{\beta} + c, \tag{19}
$$

*where* $\mathbf{P} \in \mathbb{R}^{d_0 \times (\sum_{i=1}^{L-1} d_i)}$ *is a matrix containing blocks* $\mathbf{P} := \begin{bmatrix} \mathbf{P}_1^\top & \mathbf{P}_2^\top & \cdots & \mathbf{P}_{L-1}^\top \end{bmatrix}$, $\mathbf{q} \in \mathbb{R}^{\sum_{i=1}^{L-1} d_i}$ *is a vector* $\mathbf{q} := \begin{bmatrix} \mathbf{q}_1^\top & \cdots & \mathbf{q}_{L-1}^\top \end{bmatrix}^\top$, *and each term is defined as:*

$$
\boldsymbol{a} = \left[ \boldsymbol{\Omega}(L, 1) \mathbf{W}^{(1)} \right]^\top \in \mathbb{R}^{d_0 \times 1} \tag{20}
$$

$$
\mathbf{P}_i = \mathbf{S}^{(i)} \boldsymbol{\Omega}(i, 1) \mathbf{W}^{(1)} \in \mathbb{R}^{d_i \times d_0}, \quad \forall 1 \leq i \leq L-1 \tag{21}
$$

$$
\mathbf{q}_i = \sum_{k=1}^i \mathbf{S}^{(i)} \boldsymbol{\Omega}(i, k) \mathbf{b}^{(k)} + \sum_{k=2}^i \mathbf{S}^{(i)} \boldsymbol{\Omega}(i, k) \mathbf{W}^{(k)} \underline{\mathbf{b}}^{(k-1)} \in \mathbb{R}^{d_i}, \quad \forall 1 \leq i \leq L-1 \tag{22}
$$

$$
c = \sum_{i=1}^L \boldsymbol{\Omega}(L, i) \mathbf{b}^{(i)} + \sum_{i=2}^L \boldsymbol{\Omega}(L, i) \mathbf{W}^{(i)} \underline{\mathbf{b}}^{(i-1)} \tag{23}
$$

*diagonal matrices* $\mathbf{D}^{(i)}$ *and vector* $\underline{\mathbf{b}}^{(i)}$ *are determined by the relaxation of ReLU neurons, and* $\mathbf{A}^{(i)} \in \mathbb{R}^{1 \times d_i}$ *represents the linear relationship between* $f(x)$ *and* $\hat{z}^{(i)}$. $\mathbf{D}^{(i)}$ *and* $\underline{\mathbf{b}}^{(i)}$ *depend on* $\mathbf{A}^{(i)}$, $\mathbf{l}^{(i)}$ *and* $\mathbf{u}^{(i)}$:

$$
\mathbf{D}_{j,j}^{(i)} = \begin{cases} 1, & \text{if } \mathbf{l}_j^{(i)} \geq 0 \text{ or } j \in \mathcal{Z}^{+(i)} \\ 0, & \text{if } \mathbf{u}_j^{(i)} \leq 0 \text{ or } j \in \mathcal{Z}^{-(i)} \\ \boldsymbol{\alpha}_j, & \text{if } \mathbf{u}_j^{(i)} > 0 > \mathbf{l}_j^{(i)} \text{ and } j \notin \mathcal{Z}^{+(i)} \cup \mathcal{Z}^{-(i)} \text{ and } \mathbf{A}_{1,j}^{(i)} \geq 0 \\ \frac{\mathbf{u}_j}{\mathbf{u}_j - \mathbf{l}_j}, & \text{if } \mathbf{u}_j^{(i)} > 0 > \mathbf{l}_j^{(i)} \text{ and } j \notin \mathcal{Z}^{+(i)} \cup \mathcal{Z}^{-(i)} \text{ and } \mathbf{A}_{1,j}^{(i)} < 0 \end{cases} \tag{24}
$$

$$\underline{\mathbf{b}}_j^{(i)} = \begin{cases} 0, & \text{if } \mathbf{l}_j^{(i)} > 0 \text{ or } \mathbf{u}_j^{(i)} \leq 0 \text{ or } j \in \mathcal{Z}^{+(i)} \cup \mathcal{Z}^{-(i)} \\ 0, & \text{if } \mathbf{u}_j^{(i)} > 0 > \mathbf{l}_j^{(i)} \text{ and } j \notin \mathcal{Z}^{+(i)} \cup \mathcal{Z}^{-(i)} \text{ and } \mathbf{A}_{1,j}^{(i)} \geq 0 \\ -\frac{\mathbf{u}_j^{(i)}\mathbf{l}_j^{(i)}}{\mathbf{u}_j^{(i)}-\mathbf{l}_j^{(i)}}, & \text{if } \mathbf{u}_j^{(i)} > 0 > \mathbf{l}_j^{(i)} \text{ and } j \notin \mathcal{Z}^{+(i)} \cup \mathcal{Z}^{-(i)} \text{ and } \mathbf{A}_{1,j}^{(i)} < 0 \end{cases} \tag{25}$$

$$\mathbf{A}^{(i)} = \begin{cases} \mathbf{W}^{(L)}, & i = L-1 \\ (\mathbf{A}^{(i+1)}\mathbf{D}^{(i+1)} + \boldsymbol{\beta}^{(i+1)\top}\mathbf{S}^{(i+1)})\mathbf{W}^{(i+1)}, & 0 \leq i \leq L-2 \end{cases} \tag{26}$$

*Proof.* We prove this theorem by induction: assuming we know the bounds with respect to layer $\hat{z}^{(m)}$, we derive bounds for $\hat{z}^{(m-1)}$ until we reach $m = 0$ and by definition $\hat{z}^{(0)} = x$. We first define a set of matrices and vectors $\boldsymbol{a}^{(m)}, \mathbf{P}^{(m)}, \mathbf{q}^{(m)}$, $c^{(m)}$, where $\mathbf{P}^{(m)} \in \mathbb{R}^{d_m \times (\sum_{i=m+1}^{L-1} d_i)}$ is a matrix containing blocks $\mathbf{P} := \begin{bmatrix} \mathbf{P}_{m+1}^{(m)\top} & \cdots & \mathbf{P}_{L-1}^{(m)\top} \end{bmatrix}$, $\mathbf{q} \in \mathbb{R}^{\sum_{i=m+1}^{L-1} d_i}$ is a vector $\mathbf{q} := \begin{bmatrix} \mathbf{q}_{m+1}^{(m)\top} & \cdots & \mathbf{q}_{L-1}^{(m)\top} \end{bmatrix}^{\top}$, and each term is defined as:

$$\boldsymbol{a}^{(m)} = \begin{bmatrix} \boldsymbol{\Omega}(L, m+1)\mathbf{W}^{(m+1)} \end{bmatrix}^{\top} \in \mathbb{R}^{d_m \times 1} \tag{27}$$

$$\mathbf{P}_i^{(m)} = \mathbf{S}^{(i)}\boldsymbol{\Omega}(i, m+1)\mathbf{W}^{(m+1)} \in \mathbb{R}^{d_i \times d_m}, \quad \forall\, m+1 \leq i \leq L-1 \tag{28}$$

$$\mathbf{q}_i^{(m)} = \sum_{k=m+1}^{i} \mathbf{S}^{(i)}\boldsymbol{\Omega}(i, k)\mathbf{b}^{(k)} + \sum_{k=m+2}^{i} \mathbf{S}^{(i)}\boldsymbol{\Omega}(i, k)\mathbf{W}^{(k)}\underline{\mathbf{b}}^{(k-1)} \in \mathbb{R}^{d_m}, \quad \forall\, m+1 \leq i \leq L-1 \tag{29}$$

$$c^{(m)} = \sum_{i=m+1}^{L} \boldsymbol{\Omega}(L, i)\mathbf{b}^{(i)} + \sum_{i=m+2}^{L} \boldsymbol{\Omega}(L, i)\mathbf{W}^{(i)}\underline{\mathbf{b}}^{(i-1)} \tag{30}$$

and we claim that

$$\min_{\substack{x \in \mathcal{C} \\ z \in \mathcal{Z}}} f(x) \geq \max_{\tilde{\boldsymbol{\beta}}^{(m+1)} \geq 0} \min_{\substack{x \in \mathcal{C} \\ \tilde{z}^{(m)} \in \tilde{\mathcal{Z}}^{(m)}}} (\boldsymbol{a}^{(m)} + \mathbf{P}^{(m)}\tilde{\boldsymbol{\beta}}^{(m+1)})^{\top}\hat{z}^{(m)} + \mathbf{q}^{(m)\top}\tilde{\boldsymbol{\beta}}^{(m+1)} + c^{(m)} \tag{31}$$

where $\tilde{\boldsymbol{\beta}}^{(m+1)} := \begin{bmatrix} \boldsymbol{\beta}^{(m+1)\top} \cdots \boldsymbol{\beta}^{(L-1)\top} \end{bmatrix}^{\top}$ concatenating all $\boldsymbol{\beta}^{(i)}$ variables up to layer $m+1$.

For the base case $m = L-1$, we simply have

$$\min_{x \in \mathcal{C}, z \in \mathcal{Z}} f(x) = \min_{x \in \mathcal{C}, z \in \mathcal{Z}} \mathbf{W}^{(L)}\hat{z}^{(L-1)} + \mathbf{b}^{(L)}.$$

No maximization is needed and $\boldsymbol{a}^{(m)} = \begin{bmatrix} \boldsymbol{\Omega}(L, L)\mathbf{W}^{(L)} \end{bmatrix}^{\top} = \mathbf{W}^{(L)\top}$, $c^{(m)} = \sum_{i=L}^{L} \boldsymbol{\Omega}(L, i)\mathbf{b}^{(i)} = \mathbf{b}^{(L)}$. Other terms are zero.

In Section B.1 we have shown the intuition of the proof by demonstrating how to derive the bounds from layer $\hat{z}^{(L-1)}$ to $\hat{z}^{(L-2)}$. The case for $m = L-2$ is presented in Eq. 10.

Now we show the induction from $\hat{z}^{(m)}$ to $\hat{z}^{(m-1)}$. Starting from Eq. 31, since $\hat{z}^{(m)} = \text{ReLU}(z^{(m)})$ we apply Lemma A.1 by setting $w = \begin{bmatrix} \boldsymbol{a}^{(m)} + \mathbf{P}^{(m)}\tilde{\boldsymbol{\beta}}^{(m+1)} \end{bmatrix}^{\top} := \mathbf{A}^{(m)}$. It is easy to show that $\mathbf{A}^{(m)}$ can also be equivalently and recursively defined in Eq. 26 (see Lemma C.2). Based on Lemma A.1 we have $\mathbf{D}^{(m)}$ and $\underline{\mathbf{b}}^{(m)}$ defined as in Eq. 24 and Eq. 25, so Eq. 31 becomes

$$\min_{\substack{x \in \mathcal{C} \\ z \in \mathcal{Z}}} f(x) \geq \max_{\tilde{\boldsymbol{\beta}}^{(m+1)} \geq 0} \min_{\substack{x \in \mathcal{C} \\ \tilde{z}^{(m)} \in \tilde{\mathcal{Z}}^{(m)}}} (\boldsymbol{a}^{(m)} + \mathbf{P}^{(m)}\tilde{\boldsymbol{\beta}}^{(m+1)})^{\top}\mathbf{D}^{(m)}z^{(m)}$$
$$+ (\boldsymbol{a}^{(m)} + \mathbf{P}^{(m)}\tilde{\boldsymbol{\beta}}^{(m+1)})^{\top}\underline{\mathbf{b}}^{(m)} + \mathbf{q}^{(m)\top}\tilde{\boldsymbol{\beta}}^{(m+1)} + c^{(m)} \tag{32}$$

Note that when we apply Lemma A.1, for $j \in \mathcal{Z}^{+(i)}$ (positive split) we simply treat the neuron $j$ as if $\mathbf{l}_j^{(i)} \geq 0$, and for $j \in \mathcal{Z}^{-(i)}$ (negative split) we simply treat the neuron $j$ as if $\mathbf{u}_j^{(i)} \leq 0$. Now we add the multiplier $\beta^{(m)}$ to $z^{(m)}$ to enforce per-neuron split constraints:

$$
\begin{aligned}
\min_{\substack{x \in \mathcal{C} \\ z \in \mathcal{Z}}} f(x) \geq & \max_{\tilde{\beta}^{(m+1)} \geq 0} \min_{\substack{x \in \mathcal{C} \\ \hat{z}^{(m-1)} \in \tilde{\mathcal{Z}}^{(m-1)}}} \max_{\beta^{(m)} \geq 0} \quad (a^{(m)} + \mathbf{P}^{(m)}\tilde{\beta}^{(m+1)})^\top \mathbf{D}^{(m)} z^{(m)} + \beta^{(m)\top} \mathbf{S}^{(m)} z^{(m)} \\
& \qquad\qquad\qquad\qquad\qquad + (a^{(m)} + \mathbf{P}^{(m)}\tilde{\beta}^{(m+1)})^\top \underline{\mathbf{b}}^{(m)} + \mathbf{q}^{(m)\top}\tilde{\beta}^{(m+1)} + c^{(m)} \\
\geq & \max_{\tilde{\beta}^{(m)} \geq 0} \min_{\substack{x \in \mathcal{C} \\ \hat{z}^{(m-1)} \in \tilde{\mathcal{Z}}^{(m-1)}}} \quad (a^{(m)\top} \mathbf{D}^{(m)} + \tilde{\beta}^{(m+1)\top} \mathbf{P}^{(m)\top} \mathbf{D}^{(m)} + \beta^{(m)\top} \mathbf{S}^{(m)}) z^{(m)} \\
& \qquad\qquad\qquad\qquad\qquad + (a^{(m)} + \mathbf{P}^{(m)}\tilde{\beta}^{(m+1)})^\top \underline{\mathbf{b}}^{(m)} + \mathbf{q}^{(m)\top}\tilde{\beta}^{(m+1)} + c^{(m)}
\end{aligned}
$$

Similar to what we did in Eq. 9, we swap the min and max in the second inequality due to weak duality, such that every maximization on $\beta^{(i)}$ is before min. Then, we substitute $\hat{z}^{(m)} = \mathbf{W}^{(m)}\hat{z}^{(m-1)} + \mathbf{b}^{(m)}$ and obtain:

$$
\begin{aligned}
\min_{\substack{x \in \mathcal{C} \\ z \in \mathcal{Z}}} f(x) \geq & \max_{\tilde{\beta}^{(m)} \geq 0} \min_{\substack{x \in \mathcal{C} \\ \hat{z}^{(m-1)} \in \tilde{\mathcal{Z}}^{(m-1)}}} (a^{(m)\top} \mathbf{D}^{(m)} + \tilde{\beta}^{(m+1)\top} \mathbf{P}^{(m)\top} \mathbf{D}^{(m)} + \beta^{(m)\top} \mathbf{S}^{(m)})^\top \mathbf{W}^{(m)} \hat{z}^{(m-1)} \\
& + (a^{(m)\top} \mathbf{D}^{(m)} + \tilde{\beta}^{(m+1)\top} \mathbf{P}^{(m)\top} \mathbf{D}^{(m)} + \beta^{(m)\top} \mathbf{S}^{(m)})^\top \mathbf{b}^{(m)} \\
& + (a^{(m)} + \mathbf{P}^{(m)}\tilde{\beta}^{(m+1)})^\top \underline{\mathbf{b}}^{(m)} + \mathbf{q}^{(m)\top}\tilde{\beta}^{(m+1)} + c^{(m)} \\
= & \left[ \underbrace{\left[a^{(m)\top} \mathbf{D}^{(m)} \mathbf{W}^{(m)}\right]^\top}_{a'} + \underbrace{(\tilde{\beta}^{(m+1)\top} \mathbf{P}^{(m)\top} \mathbf{D}^{(m)} \mathbf{W}^{(m)} + \beta^{(m)\top} \mathbf{S}^{(m)} \mathbf{W}^{(m)})}_{\mathbf{P}'\tilde{\beta}^{(m)}} \right]^\top \hat{z}^{(m-1)} \\
& + \underbrace{\left( (\mathbf{P}^{(m)\top} \mathbf{D}^{(m)} \mathbf{b}^{(m)} + \mathbf{P}^{(m)\top} \underline{\mathbf{b}}^{(m)} + \mathbf{q}^{(m)})^\top \tilde{\beta}^{(m+1)} + (\mathbf{S}^{(m)} \mathbf{b}^{(m)})^\top \beta^{(m)} \right)}_{\mathbf{q}'^\top \tilde{\beta}^{(m)}} \\
& + \underbrace{a^{(m)\top} \mathbf{D}^{(m)} \mathbf{b}^{(m)} + a^{(m)\top} \underline{\mathbf{b}}^{(m)} + c^{(m)}}_{c'}
\end{aligned}
$$

Now we evaluate each term $a'$, $\mathbf{P}'$, $\mathbf{q}'$ and $c'$ and show the induction holds. For $a'$ and $\mathbf{q}'$ we have:

$$
a' = \left[a^{(m)\top} \mathbf{D}^{(m)} \mathbf{W}^{(m)}\right]^\top = \left[\mathbf{\Omega}(L, m+1) \mathbf{W}^{(m+1)} \mathbf{D}^{(m)} \mathbf{W}^{(m)}\right]^\top = \left[\mathbf{\Omega}(L, m) \mathbf{W}^{(m)}\right]^\top = a^{(m-1)}
$$

$$
\begin{aligned}
c' &= c^{(m)} + \mathbf{\Omega}(L, m+1) \mathbf{W}^{(m+1)} \mathbf{D}^{(m)} \mathbf{b}^{(m)} + \mathbf{\Omega}(L, m+1) \mathbf{W}^{(m+1)} \underline{\mathbf{b}}^{(m)} \\
&= \sum_{i=m+1}^{L} \mathbf{\Omega}(L, i) \mathbf{b}^{(i)} + \sum_{i=m+2}^{L} \mathbf{\Omega}(L, i) \mathbf{W}^{(i)} \underline{\mathbf{b}}^{(i-1)} + \mathbf{\Omega}(L, m) \mathbf{b}^{(m)} + \mathbf{\Omega}(L, m+1) \mathbf{W}^{(m+1)} \underline{\mathbf{b}}^{(m)} \\
&= \sum_{i=m}^{L} \mathbf{\Omega}(L, i) \mathbf{b}^{(i)} + \sum_{i=m+1}^{L} \mathbf{\Omega}(L, i) \mathbf{W}^{(i)} \underline{\mathbf{b}}^{(i-1)} \\
&= c^{(m-1)}
\end{aligned}
$$

For $\mathbf{P}' := \left[\mathbf{P}'_m{}^\top \;\; \cdots \;\; \mathbf{P}'_{L-1}{}^\top\right]$, we have a new block $\mathbf{P}'_m$ where

$$
\mathbf{P}'_m = \mathbf{S}^{(m)} \mathbf{W}^{(m)} = \mathbf{S}^{(m)} \mathbf{\Omega}(m, m) \mathbf{W}^{(m)} = \mathbf{P}_m^{(m-1)}
$$

for other blocks where $m + 1 \leq i \leq L - 1$,

$$\mathbf{P}'_i = \mathbf{P}^{(m)}_i \mathbf{D}^{(m)} \mathbf{W}^{(m)} = \mathbf{S}^{(i)} \mathbf{\Omega}(i, m+1) \mathbf{W}^{(m+1)} \mathbf{D}^{(m)} \mathbf{W}^{(m)} = \mathbf{S}^{(i)} \mathbf{\Omega}(i, m) \mathbf{W}^{(m)} = \mathbf{P}^{(m-1)}_i$$

For $\mathbf{q}' := \begin{bmatrix} \mathbf{q}'_m{}^\top & \cdots & \mathbf{q}'_{L-1}{}^\top \end{bmatrix}$, we have a new block $\mathbf{q}'_m$ where

$$\mathbf{q}'_m = \mathbf{S}^{(m)} \mathbf{b}^{(m)} = \sum_{k=m}^{m} \mathbf{S}^{(i)} \mathbf{\Omega}(i, k) \mathbf{b}^{(i)} = \mathbf{q}^{(m-1)}_m$$

for other blocks where $m + 1 \leq i \leq L - 1$,

$$\begin{aligned}
\mathbf{q}'_i &= \mathbf{q}^{(m)}_i + \mathbf{P}^{(m)}{}^\top \mathbf{D}^{(m)} \mathbf{b}^{(m)} + \mathbf{P}^{(m)}{}^\top \underline{\mathbf{b}}^{(m)} \\
&= \sum_{k=m+1}^{i} \mathbf{S}^{(i)} \mathbf{\Omega}(i, k) \mathbf{b}^{(k)} + \sum_{k=m+2}^{i} \mathbf{S}^{(i)} \mathbf{\Omega}(i, k) \mathbf{W}^{(k)} \underline{\mathbf{b}}^{(k-1)} + \mathbf{P}^{(m)}{}^\top \mathbf{D}^{(m)} \mathbf{b}^{(m)} + \mathbf{P}^{(m)}{}^\top \underline{\mathbf{b}}^{(m)} \\
&= \sum_{k=m+1}^{i} \mathbf{S}^{(i)} \mathbf{\Omega}(i, k) \mathbf{b}^{(k)} + \sum_{k=m+2}^{i} \mathbf{S}^{(i)} \mathbf{\Omega}(i, k) \mathbf{W}^{(k)} \underline{\mathbf{b}}^{(k-1)} \\
&\quad + \mathbf{S}^{(i)} \mathbf{\Omega}(i, m+1) \mathbf{W}^{(m+1)} \mathbf{D}^{(m)} \mathbf{b}^{(m)} + \mathbf{S}^{(i)} \mathbf{\Omega}(i, m+1) \mathbf{W}^{(m+1)} \underline{\mathbf{b}}^{(k)} \\
&= \sum_{k=m+1}^{i} \mathbf{S}^{(i)} \mathbf{\Omega}(i, k) \mathbf{b}^{(k)} + \sum_{k=m+2}^{i} \mathbf{S}^{(i)} \mathbf{\Omega}(i, k) \mathbf{W}^{(k)} \underline{\mathbf{b}}^{(k-1)} + \mathbf{S}^{(i)} \mathbf{\Omega}(i, m) \mathbf{b}^{(m)} \\
&\quad + \mathbf{S}^{(i)} \mathbf{\Omega}(i, m+1) \mathbf{W}^{(m+1)} \underline{\mathbf{b}}^{(m)} \\
&= \sum_{k=m}^{i} \mathbf{S}^{(i)} \mathbf{\Omega}(i, k) \mathbf{b}^{(k)} + \sum_{k=m+1}^{i} \mathbf{S}^{(i)} \mathbf{\Omega}(i, k) \mathbf{W}^{(k)} \underline{\mathbf{b}}^{(k-1)} \\
&= \mathbf{q}^{(m-1)}_i
\end{aligned}$$

Thus, $\boldsymbol{a}' = \boldsymbol{a}^{(m-1)}$, $\mathbf{P}' = \mathbf{P}^{(m-1)}$, $\mathbf{q}' = \mathbf{q}^{(m-1)}$ and $c' = c^{(m-1)}$ so the induction holds for layer $\hat{z}^{(m-1)}$:

$$\min_{\substack{x \in \mathcal{C} \\ z \in \mathcal{Z}}} f(x) \geq \max_{\tilde{\boldsymbol{\beta}}^{(m)} \geq 0} \min_{\substack{x \in \mathcal{C} \\ \hat{z}^{(m-1)} \in \hat{\mathcal{Z}}^{(m-1)}}} (\boldsymbol{a}^{(m-1)} + \mathbf{P}^{(m-1)} \tilde{\boldsymbol{\beta}}^{(m)})^\top \hat{z}^{(m-1)} + \mathbf{q}^{(m-1)\top} \tilde{\boldsymbol{\beta}}^{(m)} + c^{(m-1)} \tag{33}$$

Finally, Theorem B.1 becomes the special case where $m = 0$ in Eq. 27, Eq. 28, Eq. 29 and Eq. 30. $\qquad\square$

The next Lemma unveils the connection with CROWN (Zhang et al., 2018) and is also useful for drawing connections to the dual problem.

**Lemma C.2.** *With* $\mathbf{D}$*,* $\underline{\mathbf{b}}$ *and* $\mathbf{A}$ *defined in Eq. 24, Eq. 25 and Eq. 26, we can rewrite Eq. 19 in Theorem B.1 as:*

$$\min_{\substack{x \in \mathcal{C} \\ z \in \mathcal{Z}}} f(x) \geq \max_{\boldsymbol{\beta} \geq 0} \min_{x \in \mathcal{C}} \mathbf{A}^{(0)} x + \sum_{i=1}^{L-1} \mathbf{A}^{(i)} (\mathbf{D}^{(i)} \mathbf{b}^{(i)} + \underline{\mathbf{b}}^{(i)}) \tag{34}$$

*where* $\mathbf{A}^{(i)}$*,* $0 \leq i \leq L - 1$ *contains variables* $\boldsymbol{\beta}$*.*

*Proof.* To prove this lemma, we simply follow the definition of $\mathbf{A}^{(i)}$ and check the resulting terms are the same as Eq. 19. For example,

$$\mathbf{A}^{(0)} = (\mathbf{A}^{(1)}\mathbf{D}^{(1)} + \boldsymbol{\beta}^{(1)\top}\mathbf{S}^{(1)})\mathbf{W}^{(1)}$$

$$= \mathbf{A}^{(1)}\mathbf{D}^{(1)}\mathbf{W}^{(1)} + \boldsymbol{\beta}^{(1)\top}\mathbf{S}^{(1)}\mathbf{W}^{(1)}$$

$$= (\mathbf{A}^{(2)}\mathbf{D}^{(2)} + \boldsymbol{\beta}^{(2)\top}\mathbf{S}^{(2)})\mathbf{W}^{(2)}\mathbf{D}^{(1)}\mathbf{W}^{(1)} + \boldsymbol{\beta}^{(1)\top}\mathbf{S}^{(1)}\mathbf{W}^{(1)}$$

$$= \mathbf{A}^{(2)}\mathbf{D}^{(2)}\mathbf{W}^{(2)}\mathbf{D}^{(1)}\mathbf{W}^{(1)} + \boldsymbol{\beta}^{(2)\top}\mathbf{S}^{(2)}\mathbf{W}^{(2)}\mathbf{D}^{(1)}\mathbf{W}^{(1)} + \boldsymbol{\beta}^{(1)\top}\mathbf{S}^{(1)}\mathbf{W}^{(1)}$$

$$= \cdots$$

$$= \boldsymbol{\Omega}(L,1)\mathbf{W}^{(1)} + \sum_{i=1}^{L-1} \boldsymbol{\beta}^{(i)\top}\mathbf{S}^{(i)}\boldsymbol{\Omega}(i,1)\mathbf{W}^{(1)}$$

$$= [\boldsymbol{a} + \mathbf{P}\boldsymbol{\beta}]^{\top}$$

Other terms can be shown similarly.

$\square$

With this definition of $\mathbf{A}$, we can see Eq. 19 as a modified form of CROWN, with an extra term $\boldsymbol{\beta}^{(i+1)\top}\mathbf{S}^{(i+1)}$ added when computing $\mathbf{A}^{(i)}$. When we set $\boldsymbol{\beta} = 0$, we obtain the same bound propagation rule for $\mathbf{A}$ as in CROWN. Thus, only a small change is needed to implement $\beta$-CROWN given an existing CROWN implementation: we add $\boldsymbol{\beta}^{(i+1)\top}\mathbf{S}^{(i+1)}$ after the linear bound propagates backwards through a ReLU layer. We also have the same observation in the dual space, as we will show this connection in the next subsection.

### C.2. Proofs for the connection to the dual space

**Theorem B.2.** *The objective $d_{LP}$ for the dual problem of Eq. 14 can be represented as*

$$d_{LP} = -\|\boldsymbol{a} + \mathbf{P}\boldsymbol{\beta}\|_1 \cdot \epsilon + (\mathbf{P}^\top x_0 + \mathbf{q})^\top \boldsymbol{\beta} + \boldsymbol{a}^\top x_0 + c,$$

*where $\boldsymbol{a}$, $\mathbf{P}$, $\mathbf{q}$ and $c$ are defined in the same way as in Theorem B.1, and $\boldsymbol{\beta} \geq 0$ corresponds to the dual variables of neuron split constraints in Eq. 14.*

*Proof.* To prove the Theorem B.2, we demonstrate the detailed dual objective $d_{LP}$ for Eq. 14, following a construction similar to the one in Wong & Kolter (2018). We first associate a dual variable for each constraint involved in Eq. 14 including dual variables $\boldsymbol{\beta}$ for the per-neuron split constraints introduced by BaB. Although it is possible to directly write the dual LP for Eq. 14, for easier understanding, we first rewrite the original primal verification problem into its Lagrangian dual form as Eq. 35, with dual variables $\boldsymbol{\nu}, \boldsymbol{\xi}^+, \boldsymbol{\xi}^-\boldsymbol{\mu}, \boldsymbol{\gamma}, \boldsymbol{\lambda}, \boldsymbol{\beta}$:

$$L(z, \hat{z}; \boldsymbol{\nu}, \boldsymbol{\xi}, \boldsymbol{\mu}, \boldsymbol{\gamma}, \boldsymbol{\lambda}, \boldsymbol{\beta}) = z^{(L)} + \sum_{i=1}^{L} \boldsymbol{\nu}^{(i)\top}(z^{(i)} - \mathbf{W}^{(i)}\hat{z}^{(i-1)} - \mathbf{b}^{(i)})$$

$$+ \boldsymbol{\xi}^{+\top}(\hat{z}^{(0)} - x_0 - \epsilon) + \boldsymbol{\xi}^{-\top}(-\hat{z}^{(0)} + x_0 - \epsilon)$$

$$+ \sum_{i=1}^{L-1} \sum_{\substack{j \notin \mathcal{Z}^{+(i)} \bigcup \mathcal{Z}^{-(i)} \\ \mathbf{l}_j^{(i)} < 0 < \mathbf{u}_j^{(i)}}} \left[ \boldsymbol{\mu}_j^{(i)\top}(-\hat{z}_j^{(i)}) + \boldsymbol{\gamma}_j^{(i)\top}(z_j^{(i)} - \hat{z}_j^{(i)}) + \boldsymbol{\lambda}_j^{(i)\top}(-\mathbf{u}_j^{(i)}z_j^{(i)} + (\mathbf{u}_j^{(i)} - \mathbf{l}_j^{(i)})\hat{z}_j^{(i)} + \mathbf{u}_j^{(i)}\mathbf{l}_j^{(i)}) \right]$$

$$\tag{35}$$

$$+ \sum_{i=1}^{L-1} \left[ \sum_{z_j^{(i)} \in \mathcal{Z}^{-(i)}} \boldsymbol{\beta}_j^{(i)} z_j^{(i)} + \sum_{z_j^{(i)} \in \mathcal{Z}^{+(i)}} -\boldsymbol{\beta}_j^{(i)} z_j^{(i)} \right]$$

Subject to:

$$\boldsymbol{\xi}^+ \geq 0, \boldsymbol{\xi}^- \geq 0, \boldsymbol{\mu} \geq 0, \boldsymbol{\gamma} \geq 0, \boldsymbol{\lambda} \geq 0, \boldsymbol{\beta} \geq 0$$

The original minimization problem then becomes:

$$\max_{\boldsymbol{\nu},\boldsymbol{\xi}^+,\boldsymbol{\xi}^-,\boldsymbol{\mu},\boldsymbol{\gamma},\boldsymbol{\lambda},\boldsymbol{\beta}} \min_{z,\hat{z}} L(z, \hat{z}, \boldsymbol{\nu}, \boldsymbol{\xi}^+, \boldsymbol{\xi}^-, \boldsymbol{\mu}, \boldsymbol{\gamma}, \boldsymbol{\lambda}, \boldsymbol{\beta})$$

Given fixed intermediate bounds $\mathbf{l}, \mathbf{u}$, the inner minimization is a linear optimization problem and we can simply transfer it to the dual form. To further simplify the formula, we introduce notations similar to those in (Wong & Kolter, 2018), where $\hat{\boldsymbol{\nu}}^{(i-1)} = \mathbf{W}^{(i)\top}\boldsymbol{\nu}^{(i)}$ and $\boldsymbol{\alpha}_j^{(i)} = \frac{\boldsymbol{\gamma}_j^{(i)}}{\boldsymbol{\mu}_j^{(i)}+\boldsymbol{\gamma}_j^{(i)}}$. Then the dual form can be written as Eq. 36.

$$\max_{0\leq\boldsymbol{\alpha}\leq1,\boldsymbol{\beta}\geq0} g(\boldsymbol{\alpha},\boldsymbol{\beta}), \text{ where}$$

$$g(\boldsymbol{\alpha},\boldsymbol{\beta}) = -\sum_{i=1}^{L}\boldsymbol{\nu}^{(i)\top}\mathbf{b}^{(i)} - \hat{\boldsymbol{\nu}}^{(0)\top}x_0 - ||\hat{\boldsymbol{\nu}}^{(0)}||_1\cdot\epsilon + \sum_{i=1}^{L-1}\sum_{\substack{j\notin\mathcal{Z}^{+(i)}\bigcup\mathcal{Z}^{-(i)}\\ \mathbf{l}_j^{(i)}<0<\mathbf{u}_j^{(i)}}}\mathbf{l}_j^{(i)}[\boldsymbol{\nu}_j^{(i)}]^+$$

Subject to:

$$\begin{aligned}
&\boldsymbol{\nu}^{(L)} = -1, \hat{\boldsymbol{\nu}}^{(i-1)} = \mathbf{W}^{(i)\top}\boldsymbol{\nu}^{(i)}, \quad i\in\{1,\dots,L\}\\
&\boldsymbol{\nu}_j^{(i)} = 0, \quad \text{when } \mathbf{u}_j^{(i)}\leq0, i\in\{1,\dots,L-1\}\\
&\boldsymbol{\nu}_j^{(i)} = \hat{\boldsymbol{\nu}}_j^{(i)}, \quad \text{when } \mathbf{l}_j^{(i)}\geq0, i\in\{1,\dots,L-1\}\\
&\left.\begin{aligned}&[\boldsymbol{\nu}_j^{(i)}]^+ = \frac{\mathbf{u}_j^{(i)}[\hat{\boldsymbol{\nu}}_j^{(i)}]^+}{\mathbf{u}_j^{(i)}-\mathbf{l}_j^{(i)}}, [\boldsymbol{\nu}_j^{(i)}]^- = \boldsymbol{\alpha}_j^{(i)}[\hat{\boldsymbol{\nu}}_j^{(i)}]^-\\ &\boldsymbol{\lambda}_j^{(i)} = \frac{[\hat{\boldsymbol{\nu}}_j^{(i)}]^+}{\mathbf{u}_j^{(i)}-\mathbf{l}_j^{(i)}}, \boldsymbol{\alpha}_j^{(i)} = \frac{\boldsymbol{\gamma}_j^{(i)}}{\boldsymbol{\mu}_j^{(i)}+\boldsymbol{\gamma}_j^{(i)}}\end{aligned}\right\} \text{when } \mathbf{l}_j^{(i)}<0<\mathbf{u}_j^{(i)}, j\notin\mathcal{Z}^{+(i)}\bigcup\mathcal{Z}^{-(i)}, i\in\{1,\dots,L-1\}\\
&\textcolor{blue}{\boldsymbol{\nu}_j^{(i)} = -\boldsymbol{\beta}_j^{(i)}, \quad j\in\mathcal{Z}^{-(i)}, i\in\{1,\dots,L-1\}}\\
&\textcolor{blue}{\boldsymbol{\nu}_j^{(i)} = \boldsymbol{\beta}_j^{(i)} + \hat{\boldsymbol{\nu}}_j^{(i)}, \quad j\in\mathcal{Z}^{+(i)}, i\in\{1,\dots,L-1\}}\\
&\boldsymbol{\mu}\geq0, \boldsymbol{\gamma}\geq0, \boldsymbol{\lambda}\geq0, \boldsymbol{\beta}\geq0, 0\leq\boldsymbol{\alpha}\leq1
\end{aligned} \tag{36}$$

Similar to the dual form in (Wong & Kolter, 2018) (our differences are highlighted in blue), the dual problem can be viewed in the form of another deep network by backward propagating $\boldsymbol{\nu}^{(L)}$ to $\hat{\boldsymbol{\nu}}^{(0)}$ following the rules in Eq. 36. If we look closely at the conditions and coefficients when backward propagating $\boldsymbol{\nu}_j^{(i)}$ for $j$-th ReLU at layer $i$ in Eq. 36, we can observe that they match exactly to the propagation of diagonal matrices $\mathbf{D}^{(i)}, \mathbf{S}^{(i)}$, and vector $\underline{\mathbf{b}}^{(i)}$ defined in Eq. 24 and Eq. 25. Therefore, using notations in Eq. 24 and Eq. 25 we can essentially simplify the dual LP problem in Eq. 36 to:

$$\boldsymbol{\nu}^{(L)} = -1, \hat{\boldsymbol{\nu}}^{(i-1)} = \mathbf{W}^{(i)\top}\boldsymbol{\nu}^{(i)}, \boldsymbol{\nu}^{(i)} = \mathbf{D}^{(i)}\hat{\boldsymbol{\nu}}^{(i)} - \boldsymbol{\beta}^{(i)}\mathbf{S}^{(i)}, i\in\{L,\cdots,1\}$$

$$\sum_{\substack{\mathbf{l}_j^{(i)}<0<\mathbf{u}_j^{(i)}\\ j\notin\mathcal{Z}^{+(i)}\bigcup\mathcal{Z}^{-(i)}}}\mathbf{l}_j^{(i)}[\boldsymbol{\nu}_j^{(i)}]^+ = -\hat{\boldsymbol{\nu}}^{(i)T}\underline{\mathbf{b}}^{(i)}, j\in\{1,\cdots,d_i\}, i\in\{L-1,\cdots,1\} \tag{37}$$

Then we prove the key claim for this proof with induction where $\boldsymbol{a}^{(m)}$ and $\mathbf{P}^{(m)}$ are defined in Eq. 27 and Eq. 28:

$$\hat{\boldsymbol{\nu}}^{(m)} = -\boldsymbol{a}^{(m)} - \mathbf{P}^{(m)}\tilde{\boldsymbol{\beta}}^{(m+1)} \tag{38}$$

When $m = L - 1$, we can have $\hat{\boldsymbol{\nu}}^{(L-1)} = -\boldsymbol{a}^{(L-1)} - \mathbf{P}^{(L-1)}\tilde{\boldsymbol{\beta}}^{(L)} = -\left[\boldsymbol{\Omega}(L,L)\mathbf{W}^{(L)}\right]^{\top} - \mathbf{0} = -\mathbf{W}^{(L)\top}$ which is true according to Eq. 37.

Now we assume that $\hat{\boldsymbol{\nu}}^{(m)} = -\boldsymbol{a}^{(m)} - \mathbf{P}^{(m)}\tilde{\boldsymbol{\beta}}^{(m+1)}$ holds, and we show that $\hat{\boldsymbol{\nu}}^{(m-1)} = -\boldsymbol{a}^{(m-1)} - \mathbf{P}^{(m-1)}\tilde{\boldsymbol{\beta}}^{(m)}$ will

hold as well:

$$\hat{\boldsymbol{\nu}}^{(m-1)} = \mathbf{W}^{(m)\top}\left(\mathbf{D}^{(m)}\hat{\boldsymbol{\nu}}^{(m)} - \boldsymbol{\beta}^{(m)}\mathbf{S}^{(m)}\right)$$

$$= -\mathbf{W}^{(m)\top}\mathbf{D}^{(m)}\boldsymbol{a}^{(m)} - \mathbf{W}^{(m)\top}\mathbf{D}^{(m)}\mathbf{P}^{(m)}\tilde{\boldsymbol{\beta}}^{(m+1)} - \mathbf{W}^{(m)\top}\boldsymbol{\beta}^{(m)}\mathbf{S}^{(m)}$$

$$= -\boldsymbol{a}^{(m-1)} - \left[\left(\mathbf{S}^{(m)}\mathbf{W}^{(m)}\right)^{\top}, \left(\mathbf{P}^{(m)\top}\mathbf{D}^{(m)}\mathbf{W}^{(m)}\right)^{\top}\right]\left[\boldsymbol{\beta}^{(m)}, \tilde{\boldsymbol{\beta}}^{(m+1)}\right]$$

$$= -\boldsymbol{a}^{(m-1)} - \mathbf{P}^{(m-1)}\tilde{\boldsymbol{\beta}}^{(m)}$$

Therefore, the claim Eq. 38 is proved with induction. Lastly, we prove the following claim where $\mathbf{q}^{(m)}$ and $c^{(m)}$ are defined in Eq. 29 and Eq. 30.

$$- \sum_{i=m+1}^{L} \boldsymbol{\nu}^{(i)\top}\mathbf{b}^{(i)} + \sum_{i=m+1}^{L-1} \sum_{\substack{\mathbf{l}_j^{(i)}<0<\mathbf{u}_j^{(i)} \\ j\notin\mathcal{Z}^{+(i)}\bigcup\mathcal{Z}^{-(i)}}} \mathbf{l}_j^{(i)}[\boldsymbol{\nu}_j^{(i)}]^{+} = \mathbf{q}^{(m)\top}\tilde{\boldsymbol{\beta}}^{(m+1)} + c^{(m)} \qquad (39)$$

This claim can be proved by applying Eq. 37 and Eq. 38.

$$- \sum_{i=m+1}^{L} \boldsymbol{\nu}^{(i)\top}\mathbf{b}^{(i)} + \sum_{i=m+1}^{L-1} \sum_{\substack{\mathbf{l}_j^{(i)}<0<\mathbf{u}_j^{(i)} \\ j\notin\mathcal{Z}^{+(i)}\bigcup\mathcal{Z}^{-(i)}}} \mathbf{l}_j^{(i)}[\boldsymbol{\nu}_j^{(i)}]^{+}$$

$$= -\sum_{i=m+1}^{L}\left(\mathbf{D}^{(i)}\hat{\boldsymbol{\nu}}^{(i)} - \boldsymbol{\beta}^{(i)}\mathbf{S}^{(i)}\right)^{\top}\mathbf{b}^{(i)} + \sum_{i=m+2}^{L}\left(-\hat{\boldsymbol{\nu}}^{(i-1)T}\underline{\mathbf{b}}^{(i-1)}\right)$$

$$= \sum_{i=m+1}^{L}\left[\left(\boldsymbol{a}^{(i)\top} + \tilde{\boldsymbol{\beta}}^{(i+1)\top}\mathbf{P}^{(i)\top}\right)\mathbf{D}^{(i)}\mathbf{b}^{(i)} + \boldsymbol{\beta}^{(i)\top}\mathbf{S}^{(i)}\mathbf{b}^{(i)}\right]$$

$$+ \sum_{i=m+2}^{L}\left(\boldsymbol{a}^{(i-1)\top} + \tilde{\boldsymbol{\beta}}^{(i)\top}\mathbf{P}^{(i-1)\top}\right)\underline{\mathbf{b}}^{(i-1)}$$

$$= \sum_{i=m+1}^{L} \tilde{\boldsymbol{\beta}}^{(i)\top}\left[\mathbf{S}^{(i)}, \mathbf{P}^{(i)\top}\mathbf{D}^{(i)}\right]\mathbf{b}^{(i)} + \sum_{i=m+2}^{L} \tilde{\boldsymbol{\beta}}^{(i)\top}\mathbf{P}^{(i-1)\top}\underline{\mathbf{b}}^{(i-1)}$$

$$+ \sum_{i=m+1}^{L} \boldsymbol{a}^{(i)\top}\mathbf{D}^{(i)}\mathbf{b}^{(i)} + \sum_{i=m+2}^{L} \boldsymbol{a}^{(i-1)\top}\underline{\mathbf{b}}^{(i-1)}$$

$$= \mathbf{q}^{(m)\top}\tilde{\boldsymbol{\beta}}^{(m+1)} + c^{(m)}$$

Finally, we apply claims Eq. 38 and Eq. 39 into the dual form solution Eq. 36 and prove the Theorem B.2.

$$g(\boldsymbol{\alpha}, \boldsymbol{\beta}) = -\sum_{i=1}^{L} \boldsymbol{\nu}^{(i)\top}\mathbf{b}^{(i)} - \hat{\boldsymbol{\nu}}^{(0)\top}x_0 - \|\hat{\boldsymbol{\nu}}^{(0)}\|_1 \cdot \epsilon + \sum_{i=1}^{L-1} \sum_{\substack{\mathbf{l}_j^{(i)}<0<\mathbf{u}_j^{(i)} \\ j\notin\mathcal{Z}^{+(i)}\bigcup\mathcal{Z}^{-(i)}}} \mathbf{l}_j^{(i)}[\boldsymbol{\nu}_j^{(i)}]^{+}$$

$$= -\|-\boldsymbol{a}^{(0)} - \mathbf{P}^{(0)}\tilde{\boldsymbol{\beta}}^{(1)}\|_1 \cdot \epsilon + \left(\boldsymbol{a}^{(0)\top} + \tilde{\boldsymbol{\beta}}^{(1)\top}\mathbf{P}^{(0)\top}\right)x_0 + \mathbf{q}^{(0)\top}\tilde{\boldsymbol{\beta}}^{(1)} + c^{(0)}$$

$$= -\|\boldsymbol{a} + \mathbf{P}\tilde{\boldsymbol{\beta}}^{(1)}\|_1 \cdot \epsilon + \left(\mathbf{P}^{\top}x_0 + \mathbf{q}\right)^{\top}\tilde{\boldsymbol{\beta}}^{(1)} + \boldsymbol{a}^{\top}x_0 + c$$

**A more intuitive proof.** Here we provide another intuitive proof showing why the dual form solution of verification objective in Eq. 36 is the same as the primal one in Thereom B.1. $d_{\text{LP}} = g(\boldsymbol{\alpha}, \boldsymbol{\beta})$ is the dual objective for Eq. 14 with free

variables $\boldsymbol{\alpha}$ and $\boldsymbol{\beta}$. We want to show that the dual problem can be viewed in the form of backward propagating $\boldsymbol{\nu}^{(L)}$ to $\hat{\boldsymbol{\nu}}^{(0)}$ following the same rules in $\beta$-CROWN. Salman et al. (2019) showed that CROWN computes the same solution as the dual form in Wong & Kolter (2018): $\hat{\boldsymbol{\nu}}^{(i)}$ is corresponding to $-\mathbf{A}^{(i)}$ in CROWN (defined in the same way as in Eq. 26 but with $\boldsymbol{\beta}^{(i+1)} = 0$) and $\boldsymbol{\nu}^{(i)}$ is corresponding to $-\mathbf{A}^{(i+1)}\mathbf{D}^{(i+1)}$. When the split constraints are introduced, extra terms for the dual variable $\boldsymbol{\beta}$ modify $\boldsymbol{\nu}^{(i)}$ (highlighted in blue in Eq. 36). The way $\beta$-CROWN modifies $\mathbf{A}^{(i+1)}\mathbf{D}^{(i+1)}$ is exactly the same as the way $\boldsymbol{\beta}^{(i)}$ affects $\boldsymbol{\nu}^{(i)}$: when we split $z_j^{(i)} \geq 0$, we add $\boldsymbol{\beta}_j^{(i)}$ to the $\boldsymbol{\nu}_j^{(i)}$ in Wong & Kolter (2018); when we split $z_j^{(i)} \geq 0$, we add $-\boldsymbol{\beta}_j^{(i)}$ to the $\boldsymbol{\nu}_j^{(i)}$ in Wong & Kolter (2018) ($\boldsymbol{\nu}_j^{(i)}$ is 0 in this case because it is set to be inactive). To make this relationship more clear, we define a new variable $\boldsymbol{\nu}'$, and rewrite relevant terms involving $\boldsymbol{\nu}, \hat{\boldsymbol{\nu}}$ below:

$$
\begin{aligned}
\boldsymbol{\nu}_j^{(i)} &= 0, \quad j \in \mathcal{Z}^{-(i)}; \\
\boldsymbol{\nu}_j^{(i)} &= \hat{\boldsymbol{\nu}}_j^{(i)}, \quad j \in \mathcal{Z}^{+(i)}; \\
\boldsymbol{\nu}_j^{(i)} &\text{ is defined in the same way as in Eq. 36 for other cases} \\
\boldsymbol{\nu}_j^{(i)\prime} &= -\boldsymbol{\beta}_j^{(i)} + \boldsymbol{\nu}_j^{(i)}, \quad j \in \mathcal{Z}^{-(i)}; \\
\boldsymbol{\nu}_j^{(i)\prime} &= \boldsymbol{\beta}_j^{(i)} + \boldsymbol{\nu}_j^{(i)}, \quad j \in \mathcal{Z}^{+(i)}; \\
\boldsymbol{\nu}_j^{(i)\prime} &= \boldsymbol{\nu}_j^{(i)}, \quad \text{otherwise} \\
\hat{\boldsymbol{\nu}}^{(i-1)} &= \mathbf{W}^{(i)\top} \boldsymbol{\nu}^{(i)\prime};
\end{aligned}
\tag{40}
$$

It is clear that $\boldsymbol{\nu}'$ corresponds to the term $-(\mathbf{A}^{(i+1)}\mathbf{D}^{(i+1)} + \boldsymbol{\beta}^{(i+1)\top}\mathbf{S}^{(i+1)})$ in Eq. 26, by noting that $\boldsymbol{\nu}^{(i)}$ in (Wong & Kolter, 2018) is equivalent to $-\mathbf{A}^{(i+1)}\mathbf{D}^{(i+1)}$ in CROWN and the choice of signs in $\mathbf{S}^{(i+1)}$ reflects neuron split constraints. Thus, the dual formulation will produce the same results as Eq. 34, and thus also equivalent to Eq. 19.

$\square$

**Corollary B.2.1.** *When $\boldsymbol{\alpha}$ and $\boldsymbol{\beta}$ are optimally set, $\beta$-CROWN produces the same solution as LP with split constraints when intermediate bounds* $\mathbf{l}, \mathbf{u}$ *are fixed. Formally,*

$$
\max_{0 \leq \boldsymbol{\alpha} \leq 1, \boldsymbol{\beta} \geq 0} g(\boldsymbol{\alpha}, \boldsymbol{\beta}) = p_{LP}^*
$$

*where $p_{LP}^*$ is the optimal objective of Eq. 14.*

*Proof.* Given fixed intermediate layer bounds $\mathbf{l}$ and $\mathbf{u}$, the dual form of the verification problem in Eq. 14 is a linear programming problem with dual variables defined in Eq. 35. Suppose we use an LP solver to obtain the optimal dual solution $\boldsymbol{\nu}^*, \boldsymbol{\xi}^*, \boldsymbol{\mu}^*, \boldsymbol{\gamma}^*, \boldsymbol{\lambda}^*, \boldsymbol{\beta}^*$. Then we can set $\alpha_j^{(i)} = \frac{\gamma_j^{(i)*}}{\mu_j^{(i)*} + \gamma_j^{(i)*}}$, $\boldsymbol{\beta} = \boldsymbol{\beta}^*$ and plug them into Eq. 36 to get the optimal dual solution $d_{\text{LP}}^*$. Theorem B.2 shows that, $\beta$-CROWN can compute the same objective $d_{\text{LP}}^*$ given the same $\alpha_j^{(i)} = \frac{\gamma_j^{(i)*}}{\mu_j^{(i)*} + \gamma_j^{(i)*}}$, $\boldsymbol{\beta} = \boldsymbol{\beta}^*$, thus $\max_{0 \leq \boldsymbol{\alpha} \leq 1, \boldsymbol{\beta} \geq 0} g(\boldsymbol{\alpha}, \boldsymbol{\beta}) \geq d_{\text{LP}}^*$. On the other hand, for any setting of $\boldsymbol{\alpha}$ and $\boldsymbol{\beta}$, $\beta$-CROWN produces the same solution $g(\boldsymbol{\alpha}, \boldsymbol{\beta})$ as the rewritten dual LP in Eq. 36, so $g(\boldsymbol{\alpha}, \boldsymbol{\beta}) \leq d_{\text{LP}}^*$. Thus, we have $\max_{0 \leq \boldsymbol{\alpha} \leq 1, \boldsymbol{\beta} \geq 0} g(\boldsymbol{\alpha}, \boldsymbol{\beta}) = d_{\text{LP}}^*$. Finally, due to the strong duality in linear programming, $p_{\text{LP}}^* = d_{\text{LP}}^* = \max_{0 \leq \boldsymbol{\alpha} \leq 1, \boldsymbol{\beta} \geq 0} g(\boldsymbol{\alpha}, \boldsymbol{\beta})$. $\square$

The variables $\boldsymbol{\alpha}$ in $\beta$-CROWN can be translated to dual variables in LP as well. Given $\boldsymbol{\alpha}^*$ in $\beta$-CROWN, we can get the corresponding dual LP variables $\boldsymbol{\mu}, \boldsymbol{\gamma}$ given $\boldsymbol{\alpha}$ by setting $\mu_j^{(i)} = (1 - \alpha_j^{(i)})[\hat{\boldsymbol{\nu}}_j^{(i)}]^-$ and $\gamma_j^{(i)} = \alpha_j^{(i)}[\hat{\boldsymbol{\nu}}_j^{(i)}]^-$.

### C.3. Proof for soundness and completeness

**Theorem B.3.** *$\beta$-CROWN with branch and bound on splitting ReLU neurons is sound and complete.*

*Proof.* **Soundness.** Branch and bound (BaB) with $\beta$-CROWN is sound because for each subdomain $\mathcal{C}_i := \{x \in \mathcal{C}, z \in \mathcal{Z}_i\}$, we apply Theorem B.1 to obtain a sound lower bound $\underline{f}_{\mathcal{C}_i}$ (the bound is valid for any $\boldsymbol{\beta} \geq 0$). The final bound returned by BaB is $\min_i \underline{f}_{\mathcal{C}_i}$ which represents the worst case over all subdomains, and is a sound lower bound for $x \in \mathcal{C} := \cup_i \mathcal{C}_i$.

---

**Algorithm 1** $\beta$-CROWN with branch and bound for complete verification. Comments are in brown.

1: **Inputs**: $f$, $\mathcal{C}$, $n$ (batch size), $\delta$ (tolerance), $\eta$ (maximum length of sub-domains)
2: $(\underline{f}, \overline{f}) \leftarrow \texttt{optimized\_beta\_CROWN}(f, [\mathcal{C}])$        ▷ Initially there is no split, so optimization is done over $\hat{\alpha}$
3: $\mathbb{P} \leftarrow [(\underline{f}, \overline{f}, \mathcal{C})]$        ▷ $\mathbb{P}$ is the set of all unverified sub-domains
4: **while** $\underline{f} < 0$ **and** $\overline{f} \geq 0$ **and** $\overline{f} - \underline{f} > \delta$ **and** $\texttt{length}(\mathbb{P}) < \eta$ **do**
5:     $(\mathcal{C}_1, \ldots, \mathcal{C}_n) \leftarrow \texttt{batch\_pick\_out}(\mathbb{P}, n)$        ▷ Pick sub-domains to split and removed them from $\mathbb{P}$
6:     $[\mathcal{C}_1^l, \mathcal{C}_1^u, \ldots, \mathcal{C}_n^l, \mathcal{C}_n^u] \leftarrow \texttt{batch\_split}(\mathcal{C}_1, \ldots, \mathcal{C}_n)$        ▷ Each $\mathcal{C}_i$ splits into two sub-domains $\mathcal{C}_i^l$ and $\mathcal{C}_i^u$
7:     $\left[\underline{f}_{\mathcal{C}_1^l}, \overline{f}_{\mathcal{C}_1^l}, \underline{f}_{\mathcal{C}_1^u}, \overline{f}_{\mathcal{C}_1^u}, \ldots, \underline{f}_{\mathcal{C}_n^l}, \overline{f}_{\mathcal{C}_n^l}, \underline{f}_{\mathcal{C}_n^u}, \overline{f}_{\mathcal{C}_n^u}\right] \leftarrow \texttt{optimized\_beta\_CROWN}(f, [\mathcal{C}_1^l, \mathcal{C}_1^u, \ldots, \mathcal{C}_n^l, \mathcal{C}_n^u])$   ▷ Compute lower and
     upper bounds by optimizing $\hat{\alpha}$ and $\hat{\beta}$ mentioned in Section B.3 in a batch
8:     $\mathbb{P} \leftarrow \mathbb{P} \bigcup \texttt{Domain\_Filter}\left([\underline{f}_{\mathcal{C}_1^l}, \overline{f}_{\mathcal{C}_1^l}, \mathcal{C}_1^l], [\underline{f}_{\mathcal{C}_1^u}, \overline{f}_{\mathcal{C}_1^u}, \mathcal{C}_1^u], \ldots, [\underline{f}_{\mathcal{C}_n^l}, \overline{f}_{\mathcal{C}_n^l}, \mathcal{C}_n^1], [\underline{f}_{\mathcal{C}_n^u}, \overline{f}_{\mathcal{C}_n^u}, \mathcal{C}_n^u]\right)$    ▷ Filter out verified
     sub-domains, insert the left domains back to $\mathbb{P}$
9:     $\underline{f} \leftarrow \min\{\underline{f}_{\mathcal{C}_i} \mid (\underline{f}_{\mathcal{C}_i}, \overline{f}_{\mathcal{C}_i}, \mathcal{C}_i) \in \mathbb{P}\}, i = 1, \ldots, n$        ▷ To ease notation, $\mathcal{C}_i$ here indicates either $\mathcal{C}_i^u$ or $\mathcal{C}_i^l$
10:     $\overline{f} \leftarrow \min\{\overline{f}_{\mathcal{C}_i} \mid (\underline{f}_{\mathcal{C}_i}, \overline{f}_{\mathcal{C}_i}, \mathcal{C}_i) \in \mathbb{P}\}, i = 1, \ldots, n$
11: **Outputs**: $\underline{f}, \overline{f}$

---

**Completeness.** To show completeness, we need to solve Eq. 5 to its global minimum. When there are $N$ unstable neurons, we have up to $2^N$ subdomains, and in each subdomain we have all unstable ReLU neurons split into one of the $z_j^{(i)} \geq 0$ or $z_j^{(i)} < 0$ case. The final solution obtained by BaB is the min over these $2^N$ subdomains. To obtain the global minimum, we must ensure that in every of these $2^N$ subdomain we can solve Eq. 14 exactly.

When all unstable neurons are split in a subdomain $\mathcal{C}_i$, the network becomes a linear network and neuron split constraints become linear constraints w.r.t. inputs. Under this case, an LP with Eq. 14 can solve the verification problem in $\mathcal{C}_i$ exactly. In $\beta$-CROWN, we solve the subdomain using the usually non-concave formulation Eq. 16; however, in this case, it becomes concave in $\hat{\beta}$ because no intermediate layer bounds are used (no $\alpha'$ and $\beta'$) and no ReLU neuron is relaxed (no $\alpha$), thus the only optimizable variable is $\beta$ (Eq. 16 becomes Eq. 12). Eq. 12 is concave in $\beta$ so (super)gradient ascent guarantees to converge to the global optimal $\beta^*$. To ensure convergence without relying on a preset learning rate, a line search can be performed in this case. Then, according to Corollary B.2.1, this optimal $\beta^*$ corresponds to the optimal dual variable for the LP in Eq. 14 and the objective is a global minimum of Eq. 14.      □

## D. More details on $\beta$-CROWN with branch and bound (BaB)

### D.1. $\beta$-CROWN with branch and bound for complete verification

We list our $\beta$-CROWN with branch and bound based complete verifier ($\beta$-CROWN BAB) in Algorithm 1. The algorithm takes a target NN function $f$ and a domain $\mathcal{C}$ as inputs. The subprocedure $\texttt{optimized\_beta\_CROWN}$ optimizes $\hat{\alpha}$ and $\hat{\beta}$ (free variables for computing intermediate layer bounds and last layer bounds) as Eq. 16 in Section B.3. It operates in a batch and returns the lower and upper bounds for $n$ selected subdomains simultaneously: a lower bound is obtained by optimizing Eq. 16 using $\beta$-CROWN and an upper bound can be the network prediction given the $x^*$ that minimizes Eq. 11[1]. Initially, we don't have any splits, so we only need to optimize $\hat{\alpha}$ to obtain $\underline{f}$ for $x \in \mathcal{C}$ (Line 2). Then we utilize the power of GPUs to split in parallel and maintain a global set $\mathbb{P}$ storing all the sub-domains which does not satisfy $\underline{f}_{\mathcal{C}_i} < 0$ (Line 5-10). Specifically, $\texttt{batch\_pick\_out}$ extends branching strategy BaBSR (Bunel et al., 2018) or FSB (De Palma et al., 2021b) in a parallel manner to select $n$ (batch size) sub-domains in $\mathbb{P}$ and determine the corresponding ReLU neuron to split for each of them. If the length of $\mathbb{P}$ is less than $n$, then we reduce $n$ to the length of $\mathbb{P}$. $\texttt{batch\_split}$ splits each selected $\mathcal{C}_i$ to two sub-domains $\mathcal{C}_i^l$ and $\mathcal{C}_i^u$ by forcing the selected unstable ReLU neuron to be positive and negative, respectively. $\texttt{Domain\_Filter}$ filters out verified sub-domains (proved with $\underline{f}_{\mathcal{C}_i} \geq 0$) and we insert the remaining ones to $\mathbb{P}$. The loop breaks if the property is proved ($\underline{f} \geq 0$), or a counter-example is found in any sub-domain ($\overline{f} < 0$), or the lower bound $\underline{f}$ and upper bound $\overline{f}$ are sufficiently close, or the length of sub-domains $\mathbb{P}$ reaches a desired threshold $\eta$ (maximum memory limit).

---

[1]We want an upper bound of the objective in Eq. 5. Since Eq. 5 is an minimization problem, any feasible $x$ produces an upper bound of the optimal objective. When Eq. 5 is solved exactly as $f^*$ (such as in the case where all neurons are split), we have $f^* = \underline{f} = \overline{f}$. See also the discussions in Section I.1 of De Palma et al. (2021a).

## D.2. Comparisons to other GPU based complete verifiers

Bunel et al. (2020a) proposed to reformulate the linear programming problem in Eq. 14 through Lagrangian decomposition. Eq. 14 is decomposed layer by layer, and each layer is solved with simple closed form solutions on GPUs. A Lagrangian is used to enforce the equality between the output of a previous layer and the input of a later layer. This optimization formulation has the same power as a LP (Eq. 14) under convergence. The main drawback of this approach is that it converges relatively slowly (we find that it typically requires hundreds of iterations to converge to a solution similar to the solution of a LP), and it also cannot easily jointly optimize intermediate layer bounds. In Table 1 (PROX BABSR) and Figure 1 (BDD+ BABSR, which refers to the same method) we can see that this approach is relatively slow and has high timeout rates compared to other GPU accelerated complete verifiers. Recently, De Palma et al. (2021b) proposed a better branching strategy, filtered smart branching (FSB), to further improved verification performance of (Bunel et al., 2020a), but the Lagrangian Decomposition based incomplete verifier and the branch and bound procedure stay the same.

De Palma et al. (2021a) used a tighter convex relaxation (Anderson et al., 2020) than the typical LP formulation in Eq. 14 for the incomplete verifier. This tighter relaxation contains exponentially many constraints, and De Palma et al. (2021a) proposed to solve the verification problem in its dual form where each constraint becomes a dual variable. A small active set of dual variables is maintained during dual optimization to ensure efficiency. This tighter relaxation allows it to outperform (Bunel et al., 2020a), but it also comes with extra computational costs and difficulties for an efficient implementation (e.g. a "masked" forward/backward pass is needed which requires a customised lower-level convolution implementation). Additionally, De Palma et al. (2021a) did not optimize intermediate layer bounds jointly.

Xu et al. (2021) used CROWN (Zhang et al., 2018) (categorized as a linear relaxation based perturbation analysis (LiRPA) algorithm) as the incomplete solver in BaB. Since CROWN cannot encode neural split constraints, Xu et al. (2021) essentially solve Eq. 14 *without* neuron split constraints ($z_j^{(i)} \geq 0, i \in \{1, \cdots, L-1\}, j \in \mathcal{Z}^{+(i)}$ and $z_j^{(i)} < 0, i \in \{1, \cdots, L-1\}, j \in \mathcal{Z}^{-(i)}$) in Eq. 14. The missing constraints lead to looser bounds and more branches - this can be seen in Table 1, where their number of branches and timeout rates are higher than ours. Additionally, using CROWN as the incomplete solver leads to incompleteness - even when all unstable ReLU neurons are split, Xu et al. (2021) still cannot solve Eq. 5 to a global minimum, so a LP solver has to be used to check inconsistent splits and guarantee completeness. Our $\beta$-CROWN BaBSR and $\beta$-CROWN FSB overcome these drawbacks: we consider per-neuron split constraints in $\beta$-CROWN which reduces the number of branches and solving time (Table 1). Most importantly, $\beta$-CROWN with branch and bound is sound and complete (Theorem B.3) and we do not rely on any LP solvers.

Another difference between Xu et al. (2021) and our method is the joint optimization of intermediate layer bounds (Section B.3). Although (Xu et al., 2021) also optimized intermediate layer bounds, they only optimize $\boldsymbol{\alpha}$ and do not have $\boldsymbol{\beta}$, and they share the same variable $\boldsymbol{\alpha}$ for all intermediate layer bounds and final bounds, with a total of $O(Ld)$ variables to optimize. Our analysis in Section B.3 shows that there are in fact, $O(L^2 d^2)$ free variables to optimize, and we share less variables as in Xu et al. (2021). This allows us to achieve tighter bounds and improve overall performance.

## D.3. Detection of Infeasibility

Maximizing Eq. 12 with infeasible constraints leads to unbounded dual objective, which can be detected by checking if this optimized lower bound becomes *greater than the upper bound* (which is also maintained in BaB, see Alg.1 in Sec. B.1). Due to insufficient convergence, this cannot always detect infeasibility, but it *does not affect soundness*, as this infeasible subdomain only leads to worse overall lower bound in BaB. To guarantee *completeness*, we show that when all unstable neurons are split the problem is concave (see Section C.3); in this case, we can use line search to guarantee convergence when feasible, and detect infeasibility if the objective exceeds the upper bound (line search guarantees the objective can eventually exceed upper bound). In most real scenarios, the verifier either finishes or times out before all unstable neurons are split.

# E. Details on Experimental Setup and Results

## E.1. Baselines

We compare against multiple baselines for complete verification: (1) BaBSR (Bunel et al., 2020b), a basic BaB and LP based verifier; (2) MIPplanet (Ehlers, 2017), a customized MIP solver for NN verification where unstable ReLU neurons are randomly selected for splitting; (3) ERAN (Singh et al., 2019a; 2018a; 2019b; 2018b), an abstract interpretation

based verifier which is one of VNN competition 2020 winners with lowest timeout rate; (4) GNN-Online (Lu & Kumar, 2020), a BaB and LP based verifiers using a learned Graph Neural Network (GNN) to guide the ReLU splits; (5) BDD+ BaBSR (Bunel et al., 2020a), a verification framework based on Lagrangian decomposition on GPUs (BDD+) with BaBSR branching strategy; (6) OVAL (BDD+ GNN) (Bunel et al., 2020a; Lu & Kumar, 2020), one of VNN competition 2020 winners, using BDD+ with GNN guiding the ReLU splits; (7) A.set BaBSR and (8) Big-M+A.set BaBSR (De Palma et al., 2021a), very recent dual-space verifiers on GPUs with a tighter linear relaxation than LP; (9) Fast-and-Complete (Xu et al., 2021), which uses CROWN on GPUs as the incomplete verifier in BaB without neuron split constraints; (10) BaDNB (BDD+ FSB) (De Palma et al., 2021b), a concurrent state-of-the-art complete verifier, using BDD+ on GPUs with FSB branching strategy. $\beta$-CROWN BaB can use either BaBSR or FSB branching heuristic, and we include both in evaluation. All methods use 1 CPU + 1 GPU (if GPU is supported) with a 1 hour timeout threshold.

## E.2. Experimental Setup

We run our experiments on a single NVIDIA GTX 1080 Ti GPU (11GB GPU memory) and a Intel i7-7700K CPU (4.2 GHz). We set the CPU memory limit for all methods to 32GB. We use the Adam optimizer (Kingma & Ba, 2015) to solve both $\hat{\alpha}$ and $\hat{\beta}$ in Eq. 16 with 20 iterations. The learning rates are set as 0.1 and 0.05 for optimizing $\hat{\alpha}$ and $\hat{\beta}$ respectively. We decay the learning rates with a factor of 0.98 per iteration. To maximize the benefits of parallel computing on GPU, we use batch sizes $n =$400 for Base (CIFAR-10), ConvSmall (MNIST), ConvSmall (CIFAR-10), CNN-A-Adv (MNIST), CNN-A-Adv (CIFAR-10), CNN-A-Adv-4 (CIFAR-10), CNN-A-Mix (CIFAR-10) and CNN-A-Mix-4 (CIFAR-10); $n =$200 for Wide (CIFAR-10) and Deep (CIFAR-10), $n =$64 for CNN-B-Adv (CIFAR-10) and CNN-B-Adv-4 (CIFAR-10), $n =$10 for ConvBig (MNIST) and ConvBig (CIFAR-10); $n =$2 for ResNet (CIFAR-10) respectively. The CNN-A-Adv, CNN-A-Adv-4, CNN-A-Mix, CNN-A-Mix-4, CNN-B-Adv and CNN-B-Adv-4 models are obtained from the authors or (Dathathri et al., 2020) and are the same as the models used in their paper. We summarize the model structures in both incomplete verification and complete verification (Base, Wide and Deep) experiments in Table 3.

*Table 3.* Model structures used in our experiments. For example, Conv(1, 16, 4) stands for a conventional layer with 1 input channel, 16 output channels and a kernel size of $4 \times 4$. Linear(1568, 100) stands for a fully connected layer with 1568 input features and 100 output features. We have ReLU activation functions between two consecutive layers.

| Model name | Model structure |
|---|---|
| CNN-A-Adv (MNIST) | Conv(1, 16, 4) - Conv(16, 32, 4) - Linear(1568, 100) - Linear(100, 10) |
| ConvSmall (MNIST) | Conv(1, 16, 4) - Conv(16, 32, 4) - Linear(800, 100) - Linear(100, 10) |
| ConvBig (MNIST) | Conv(1, 32, 3) - Conv(32, 32, 4) - Conv(32, 64, 3) - Conv(64, 64, 4) - Linear(3136, 512) - Linear(512, 512) - Linear(512, 10) |
| ConvSmall (CIFAR-10) | Conv(3, 16, 4) - Conv(16, 32, 4) - Linear(1152, 100) - Linear(100, 10) |
| ConvBig (CIFAR-10) | Conv(3, 32, 3) - Conv(32, 32, 4) - Conv(32, 64, 3) - Conv(64, 64, 4) - Linear(4096, 512) - Linear(512, 512) - Linear(512, 10) |
| CNN-A-Adv/-4 (CIFAR-10) | Conv(3, 16, 4) - Conv(16, 32, 4) - Linear(2048, 100) - Linear(100, 10) |
| CNN-B-Adv/-4 (CIFAR-10) | Conv(3, 32, 5) - Conv(32, 128, 4) - Linear(8192, 250) - Linear(250, 10) |
| CNN-A-Mix/-4 (CIFAR-10) | Conv(3, 16, 4) - Conv(16, 32, 4) - Linear(2048, 100) - Linear(100, 10) |
| Base (CIFAR-10) | Conv(3, 8, 4) - Conv(8, 16, 4) - Linear(1024, 100) - Linear(100, 10) |
| Wide (CIFAR-10) | Conv(3, 16, 4) - Conv(16, 32, 4) - Linear(2048, 100) - Linear(100, 10) |
| Deep (CIFAR-10) | Conv(3, 8, 4) - Conv(8, 8, 3) - Conv(8, 8, 3) - Conv(8, 8, 4) - Linear(412, 100) - Linear(100, 10) |

## E.3. Additional Experiments

**Comparison to Multi-neuron Convex Relaxation based Incomplete verifiers.** In Table 4, we compare against a few representative and strong incomplete verifiers on 5 convolutional networks for MNIST and CIFAR-10 under the same set of 1000 images and perturbation $\epsilon$ as reported in (Singh et al., 2019b; Tjandraatmadja et al., 2020; Müller et al., 2021). Among the baselines, kPoly (Singh et al., 2019a), OptC2V (Tjandraatmadja et al., 2020) and PRIMA (Müller et al., 2021) utilize state-of-the-art multi-neuron linear relaxation for ReLUs and can bypass the single-neuron convex relaxation barrier (Salman et al., 2019), and are among the strongest incomplete verifiers. $\beta$-CROWN FSB achieves better verified accuracy on all five models using a similar amount of time. Some models, such as MNIST ConvBig and CIFAR ResNet, are quite challenging - the verified accuracy obtained by $\beta$-CROWN FSB is close to the upper bound found via PGD attack.

**Tightness of verification.** In Figure 2, we compare the tightness of verification bounds against SDP-FO on two adversarially trained networks from (Dathathri et al., 2020). Specifically, we use the verification objective $f(x) := z_y^{(L)}(x) - z_{y'}^{(L)}(x)$, where $z^{(L)}$ is the logit layer output, $y$ and $y'$ are the true label and the runner-up label. For each test image, PGD at-

*Table 4.* **Verified accuracy (%)** and avg. time (s) of 1000 images evaluated on the ERAN models in (Singh et al., 2019a; Tjandraatmadja et al., 2020; Müller et al., 2021). Three convex relaxation barrier breaking methods, kPoly (Singh et al., 2019b), OptC2V (Tjandraatmadja et al., 2020) and PRIMA (Müller et al., 2021)

| Dataset | Model | CROWN/DeepPoly* | | kPoly | | OptC2V | | PRIMA[†] | | $\beta$-CROWN FSB | | Upper bound |
|---|---|---|---|---|---|---|---|---|---|---|---|---|
| | | Verified% | Time (s) | Ver.% | Time(s) | Ver.% | Time(s) | Ver.% | Time(s) | Ver.% | Time(s) | |
| MNIST | ConvSmall | 15.8 | 3 | 34.7 | 477 | 43.6 | 55 | 59.8 | 42 | **71.6** | 46 | 74.6 |
| | ConvBig | 71.1 | 21 | 73.6 | 40 | 77.1 | 102 | 77.5 | 15 | **77.7** | 78 | 80.4 |
| CIFAR | ConvSmall | 35.9 | 4 | 39.9 | 86 | 39.8 | 105 | 44.1 | 20 | **46.3** | 18 | 48.2 |
| | ConvBig | 42.1 | 43 | 45.9 | 346 | No public code | | 46.9 | 97 | **50.3** | 55 | 61.3 |
| | ResNet | 24.3 | 12 | 24.5 | 91 | cannot run | | 24.9 | 64 | **25.1** | 42 | 29.0 |

* CROWN/DeepPoly evaluated on CPU. [†] PRIMA is a concurrent work and results are directly from (Müller et al., 2021) under the same setting.

tack (Madry et al., 2018) can provide an adversarial upper bound $\overline{f}$ of the optimal objective: $f^* \leq \overline{f}$. Verifiers, on the other hand, can provide a verified lower bound $\underline{f} \leq f^*$. Bounds from tighter verification methods lie closer to line $y = x$ in Figure 2. Figure 2 shows that on both PGD adversarially trained networks, $\beta$-CROWN FSB consistently outperforms SDP-FO for all 100 random test images. Importantly, for each point on the plots, $\beta$-CROWN FSB needs only 3 minutes while SDP-FO needs 178 minutes on average. LP verifier produces much looser bounds than $\beta$-CROWN FSB and SDP-FO.

*Figure 2.* Verified lower bound v.s. PGD adversarial upper bound. A lower bound closer to the upper bound (closer to the line $y = x$) is better. $\beta$-CROWN FSB uses 3mins while SDP-FO needs roughly 3 hours per point.

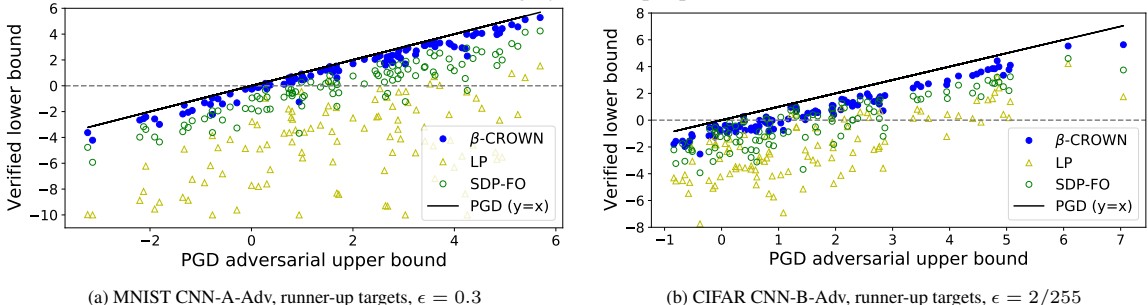

(a) MNIST CNN-A-Adv, runner-up targets, $\epsilon = 0.3$      (b) CIFAR CNN-B-Adv, runner-up targets, $\epsilon = 2/255$

**Comparison to LPs with different intermediate layer bounds.** In Figure 3, we compare our $\beta$-CROWN FSB (3 minutes as the timeout threshold per verification instance) against incomplete LP verifiers constructed using different intermediate layer bounds obtained from Wong & Kolter (2018) (WK), CROWN (Zhang et al., 2018), and the joint optimization procedure (optimizing Eq. 16 with no $\hat{\beta}$ as done in (Xu et al., 2021), denoted as OPT). Our $\beta$-CROWN FSB always outperforms these three LP verifiers using different intermediate bounds. Also, we show that tighter intermediate layer bounds obtained by CROWN can greatly improve the performance of the LP verifier compared to those using looser ones obtained by Wong & Kolter (2018). Furthermore, using intermediate layer bounds computed by joint optimization can achieve noticeable improvements. The corresponding verified accuracy for each method on PGD trained CNN-A-Adv (MNIST) and CNN-B-Adv (CIFAR-10) networks can be found in Table 5. The results match the observations in Figure 3: tighter intermediate bounds are helpful for LPs, but branch and bound with $\beta$-CROWN can significantly outperform these LP verifiers. This shows that BaB is an effective approach for incomplete verification, outperforming the bounds produced by a single LP.

**More results on incomplete verification using complete verifiers with early stop** In our experiments in Table 1, we noticed that BIGM+A.SET BABSR (De Palma et al., 2021a) is also very competitive among existing state-of-the-art complete verifiers[2] - it runs fast in many cases with low timeout rates. Therefore, we also evaluate BIGM+A.SET BABSR with an early stop of 3 minutes for the incomplete verification setting as an extension of Section 4. The corresponding verified accuracy number for each method are reported in Table 5. As we can see BIGM+A.SET BABSR usually produces better bounds than SDP-FO, however $\beta$-CROWN FSB consistently outperforms BIGM+A.SET BABSR under the same 3min timeout.

---

[2]The concurrent work BaDNB (BDD+ FSB) does not have public available code.

*Figure 3.* Verified lower bound on $f(x)$ by $\beta$-CROWN FSB compared against incomplete LP verifiers using different intermediate layer bounds obtained from (Wong & Kolter, 2018) (denoted as LP (WK)), CROWN (Zhang et al., 2018) (denoted as LP (CROWN)), and jointly optimized intermediate bounds in Eq. 16 (denoted as LP (OPT)), v.s. the adversarial upper bound on $f(x)$ found by PGD. LPs need much longer time to solve than $\beta$-CROWN on CIFAR-10 models (see Table 5).

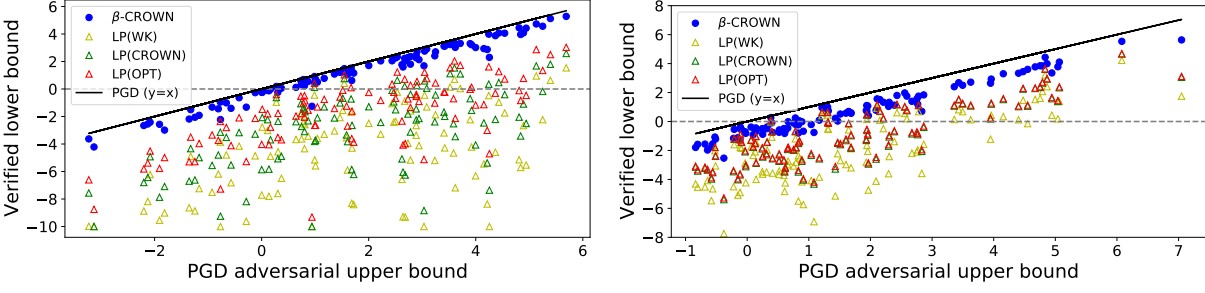

(a) MNIST CNN-A-Adv, runner-up targets, $\epsilon = 0.3$    (b) CIFAR CNN-B-Adv, runner-up targets, $\epsilon = 2/255$

*Table 5.* **Verified accuracy (%)** and avg. per-example verification time (s) on 7 models from SDP-FO (Dathathri et al., 2020).

| Dataset | MNIST $\epsilon = 0.3$ | | CIFAR $\epsilon = 2/255$ | | | | | | | | | | | |
|---|---|---|---|---|---|---|---|---|---|---|---|---|---|---|
| Model | CNN-A-Adv | | CNN-B-Adv | | CNN-B-Adv4 | | CNN-A-Adv | | CNN-A-Adv4 | | CNN-A-Mix | | CNN-A-Mix | |
| Methods | Verified% | Time (s) | Ver.% | Time(s) | Ver.% | Time(s) | Ver.% | Time(s) | Ver.% | Time(s) | Ver.% | Time(s) | Ver.% | Time(s) |
| K&W (Wong & Kolter, 2018) | 0 | 0.1 | 8.5 | 0.4 | 34.5 | 0.8 | 32.5 | 0.4 | 39.5 | 0.5 | 15.0 | 0.3 | 30.0 | 0.4 |
| CROWN (Zhang et al., 2018) | 1.0 | 0.1 | 21.5 | 0.5 | 43.5 | 0.9 | 35.5 | 0.6 | 41.5 | 0.7 | 23.5 | 0.4 | 38.0 | 0.5 |
| CROWN-OPT (Xu et al., 2021) | 14.0 | 3 | 21.5 | 6 | 45.0 | 4 | 36.0 | 2 | 42.0 | 2 | 25.0 | 2 | 38.5 | 2 |
| LP(K&W) | 0.5 | 16 | 14.5 | 612 | 41.0 | 1361 | 35.0 | 114 | 41.5 | 140 | 19.0 | 84 | 36.5 | 117 |
| LP(CROWN) | 3.5 | 22 | 21.5 | 941 | 45.0 | 1570 | 36.0 | 123 | 41.5 | 147 | 24.0 | 119 | 38.5 | 126 |
| LP(OPT) | 14.0 | 40 | 21.5 | 977 | 45.0 | 1451 | 36.0 | 122 | 42.0 | 152 | 25.0 | 94.8 | 38.5 | 127 |
| SDP-FO (Dathathri et al., 2020)* | 43.4 | >20h | 32.8 | >25h | 46.0 | >25h | 39.6 | >25h | 40.0 | >25h | 39.6 | >25h | 47.8 | >25h |
| PRIMA (Müller et al., 2021) | 44.5 | 136 | 38.0 | 360 | 53.5 | 51 | 41.5 | 11 | 45.0 | 7 | 37.5‡ | 36 | 48.5 | 9 |
| BigM+A.Set (De Palma et al., 2021a) | 63.0 | 117 | N/A† | N/A | N/A | N/A | 41.0 | 79 | **46.0** | 39 | 30.0 | 122 | 47.0 | 71 |
| $\beta$-CROWN FSB | **68.0** | 76 | **44.5** | 94 | **54.0** | 52 | **43.5** | 31 | **46.0** | 4 | **41.5** | 33 | **50.5** | 8 |
| Upper Bound (PGD) | 79.5 | - | 64.0 | - | 62.5 | - | 52.0 | - | 49.5 | - | 51.5 | - | 55.0 | - |

* SDP-FO results are directly from their paper due to the very long running time. All other methods are tested on the same set of 200 random examples. † The implementation of BigM+A.Set BaBSR is not compatible with CNN-B-Adv and CNN-B-Adv4 models which have an convolution with asymmetric padding. ‡ A recent version of (Müller et al., 2021) reported better results on CNN-A-Mix, however we found that their results were produced on a selection of 100 data points, and reruning their method using the same command on 200 random examples from test set produces much worse results, as reported here.

**Lower bound improvements over time**    In Figure 4, we plot lower bound values vs. time for $\beta$-CROWN BABSR and BIGM+A.SET BABSR (one of the most competitive methods in Table 1) on the CNN-A-Adv (MNIST) model. Figure 4 shows that branch and bound can indeed quickly improve the lower bound, and our $\beta$-CROWN BABSR is consistently faster than BIGM+A.SET BABSR. In contrast, SDP-FO (Dathathri et al., 2020), which typically requires 2 to 3 hours to converge, can only provide very loose bounds during the first 3 minutes of optimization (out of the range on these figures).

Complete verification performance with averaged metrics

In Section 4 we presented the median of verification time in Table 1. We include mean verification time and number of branches in Table 1. The average verification time is heavily affected by timed out examples. For example, on the Deep model, our $\beta$-CROWN with BaBSR significantly outperforms other baselines by over 10X because we have no timeout. This comparison can be misleading because two factors are mixed: the efficiency of the verifier (reflected in verification time for *most* examples) and the capability of the verifier (reflected in the capability of verifying hard examples and producing less timeouts). Instead, median runtime (with timeout rates also in consideration) and the cactus plots in Figure 1 are more appropriate means to gauge the performance of each complete verifier.

*Figure 4.* For the CNN-A-Adv (MNIST) model, we randomly select four examples from the incomplete verification experiment and plot the lower bound v.s. time (in 180 seconds) of $\beta$-CROWN BABSR and BIGM+A.SET BABSR. Larger lower bounds are better. $\beta$-CROWN BaBSR improves bound noticeably faster in all four situations.

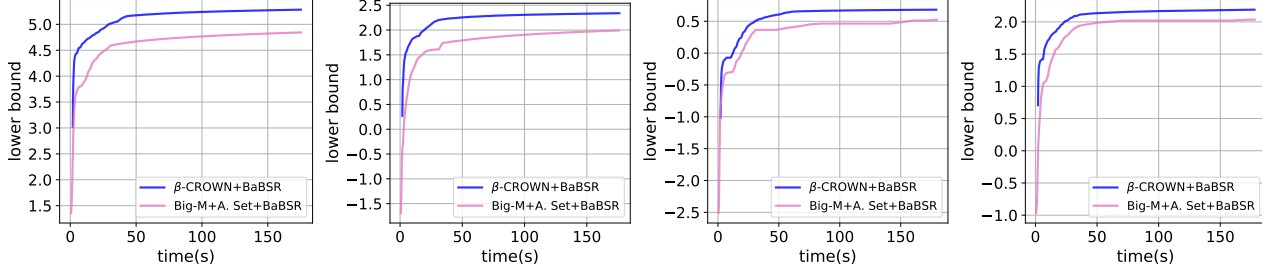