# OpenReview forum: "Beta-CROWN: Efficient Bound Propagation with Per-neuron Split Constraints for Neural Network Robustness Verification"
_ICML.cc/2021/Workshop/AML — ICML 2021 Workshop AML Poster_

### Official Review · Reviewer_dyX4 · 2021-06-19
**A novel bound propagation algorithm**

**Rating:** Accept
**Confidence:** 4

**Review:**

This paper proposed a new bound propagation algorithm named beta-CROWN. It formulates the problem equivalent to the expensive LP based methods with neuron split constraints while it is as efficient as the IBP methods. The idea is interesting and the mathematical derivation is complete. The experimental results are also impressive which is both efficient and effective.

---

### Decision · Program_Chairs · 2021-06-21

**Decision:**

Accept (Poster)

**Comment:**

This paper proposed a novel bound propagation algorithm. The idea is interesting. The experiments are solid.